# FEATURE KERNEL DISTILLATION

**Bobby He**[1,2†] **& Mete Ozay**[2]
[1]Department of Statistics, University of Oxford
[2]Samsung Research UK

## ABSTRACT

Trained Neural Networks (NNs) can be viewed as data-dependent kernel machines, with predictions determined by the inner product of last-layer representations across inputs, referred to as the *feature kernel*. We explore the relevance of the feature kernel for Knowledge Distillation (KD), using a mechanistic understanding of an NN's optimisation process. We extend the theoretical analysis of Allen-Zhu & Li (2020) to show that a trained NN's feature kernel is highly dependent on its parameter initialisation, which biases different initialisations of the same architecture to learn different data attributes in a multi-view data setting. This enables us to prove that KD using only pairwise feature kernel comparisons can improve NN test accuracy in such settings, with both single & ensemble teacher models, whereas standard training without KD fails to generalise. We further use our theory to motivate practical considerations for improving student generalisation when using distillation with feature kernels, which allows us to propose a novel approach: Feature Kernel Distillation (FKD). Finally, we experimentally corroborate our theory in the image classification setting, showing that FKD is amenable to ensemble distillation, can transfer knowledge across datasets, and outperforms both vanilla KD & other feature kernel based KD baselines across a range of standard architectures & datasets.

## 1 INTRODUCTION & BACKGROUND

A prevailing belief in the Deep Learning community is that feature learning, where data-dependent features are acquired during training, is crucial to explaining the empirical success of Neural Networks (NNs) (Fort et al., 2020; Baratin et al., 2021). A comparison in this regard is often made to kernel methods (Jacot et al., 2018), which can be thought of as feature selection methods from a *fixed* data-independent set of features. This separation has been caricaturised as a distinction between *feature learning* and *kernel learning* regimes (Chizat et al., 2019; Yang & Hu, 2020; Woodworth et al., 2020) of NN training. Though less amenable to theoretical analysis compared to kernel regimes, feature learning regimes have the capability to capture more of the complex empirical phenomenon that one can observe in NNs due to parameter-space non-convexity, such as: i) how ensembling trained NNs differing solely in their independent parameter initialisations can lead to improvements in predictive accuracy & uncertainty (Lakshminarayanan et al., 2017; Allen-Zhu & Li, 2020), or ii) the effectiveness of knowledge distillation with both single & ensemble teacher models (Buciluǎ et al., 2006; Hinton et al., 2015). This implies that in order to understand ensembling & knowledge distillation (KD) in NNs, we need to understand the mechanisms of NN feature learning regimes.

Ensembling can be loosely summarised as aggregating predictions from multiple models, & is used widely across machine learning (ML) to improve performance (Dietterich, 2000; Breiman, 2001). Conversely, knowledge distillation (KD), the idea of transferring knowledge from a teacher model to a student model, has garnered the most attention with NNs. Remarkably, via KD it is possible to

---

[†]Researched during internship at Samsung Research UK. Correspondence to bobby.he@stats.ox.ac.uk.

significantly improve a single student's generalisation with knowledge from a teacher model, or an ensemble of teachers. This means that the single student's model has enough flexibility to generalise well (relative to the teacher), thus one must factor in the optimisation process (such as the parameter initialisation) in order to explain the mechanisms of ensembling and KD in NNs.

To describe KD, suppose we have $N$ input-target $(\boldsymbol{x}, \boldsymbol{y}) \in \mathbb{R}^d \times \mathbb{R}^{C_S}$ data pairs $\hat{\mathcal{D}} = \{(\boldsymbol{x}_i, \boldsymbol{y}_i)\}_{i=1}^N$ sampled i.i.d. (independent & identically distributed) from some distribution $\mathcal{D}$, and a student NN architecture $f_S(\boldsymbol{x}) = W_S \cdot h_S(\boldsymbol{x}, \boldsymbol{\theta}_S)$. Here $h_S(\boldsymbol{x}, \boldsymbol{\theta}_S) \in \mathbb{R}^{m \times 1}$ is a student-specific feature extractor model (e.g. MLP, CNN, ResNet, or Transformer) with parameters $\boldsymbol{\theta}_S$, and $W_S \in \mathbb{R}^{C_S \times m}$ is a parameter matrix for the last layer. Assume also that we have loss: $\mathcal{L}(\boldsymbol{\theta}_S, W_S) = \frac{1}{N} \sum_{i=1}^N L(f_S(\boldsymbol{x}_i), \boldsymbol{y}_i)$ which we seek to minimise over $\boldsymbol{\theta}_S, W_S$ in the hope that $f_S$ can generalise to unseen $(\boldsymbol{x}, \boldsymbol{y})$ pairs. $L$ is typically cross-entropy in the classification setting.

Vanilla KD (Hinton et al., 2015) distils knowledge from a trained teacher network $f_T(\boldsymbol{x}) = W_T \cdot h_T(\boldsymbol{x}, \boldsymbol{\theta}_T) \in \mathbb{R}^{C_T}$ to a student by regularising student $f_S$ towards the teacher $f_T$:

$$\tilde{\mathcal{L}}(\boldsymbol{\theta}_S, W_S) = \mathcal{L}(\boldsymbol{\theta}_S, W_S) + \lambda_{\text{KD}} \frac{1}{N} \sum_{i=1}^N L\big(\frac{f_S(\boldsymbol{x}_i)}{\tau}, \frac{f_T(\boldsymbol{x}_i)}{\tau}\big), \tag{1}$$

for temperature $\tau > 0$ & regularisation $\lambda_{\text{KD}} > 0$ hyperparameters. Note, this is only valid if $C_T = C_S$.

Following Hinton et al. (2015), many methods have been proposed using different quirks of NNs to distil knowledge from teacher to student. A relevant line of work involves encouraging the student to match how similar/related the teacher views two inputs $\boldsymbol{x}, \boldsymbol{x}'$ to be (Passalis & Tefas, 2018; Tung & Mori, 2019; Park et al., 2019). These approaches have the benefit of being agnostic to teacher/student architectures & prediction spaces $C_T$ & $C_S$, but as of yet remain heuristically motivated. In this work, we explore such approaches under the more general framework of NN *feature kernel* (the kernel induced by the inner product of last-layer features $h$) learning, allowing us to provide the missing theoretical justification. Moreover, we use our theoretical insights to introduce practical improvements for FKD in Section 4, which we show outperform these previous works in Section 5.

Allen-Zhu & Li (2020) provide the first theoretical exposition of the mechanisms by which vanilla KD and ensembling improve generalisation in NNs. To this end, the authors introduce the notion of multi-view data, which is when a class in a multi-class classification problem has multiple identifying features/attributes. For example, an image of a car can be discerned by i) wheels, ii) windows, or iii) headlights. The key idea is that the NN parameter initialisation, and its random correlations with certain attributes, will bias the NN to learn only a subset of the entire set of attributes pertaining to a given class. When presented with single-view data lacking the class-identifying attribute that the NN has learnt, the NN will not generalise. For example, an NN that has learnt to classify cars based on if they have headlights will not generalise to a side-on image of a car that occludes headlights.

The implication then is that ensembling NNs works in part because independent parameter initialisations learn independent sets of attributes, so more data features will be learnt across the ensemble model. Moreover, it is argued that vanilla KD in NNs works because the features learnt by the teacher model (or models) are imparted to the student via soft teacher labels that capture ambiguity in a given data input (such as an image of a car whose headlights look like the eyes of a cat). This is fundamentally different to ensembling in strongly convex feature selection problems, such as linear or random features (Rahimi & Recht, 2007) models with $\ell_2$ regularisation. In such cases, different initialisations reach the same unique optimum, and additional noise must be added to ensure predictive diversity in the ensemble (Matthews et al., 2017). These analyses suggest that it is not possible to fully explain KD or ensembling in NNs without feature learning, thus motivating our study of *Feature Kernel Distillation*, where one performs KD on NN features directly.

**Our contributions** Feature learning can be thought of as when the *feature kernel*, induced by the inner product of last-layer representations in a NN, changes during training (Yang & Hu, 2020), and kernel learning in NNs can be thought of as when this kernel is constant. In this work, we take a

feature learning perspective of knowledge distillation (KD). We first highlight the importance of the feature kernel by viewing trained NNs as data-dependent kernel machines, & use this to motivate *Feature Kernel Distillation* (FKD). In FKD, we aim to ensure that the student's feature kernel is well suited for improved generalisation, using both the teacher's data-dependent feature kernel as well as an understanding of the student NN's optimisation process. In Section 3, we adapt the framework of Allen-Zhu & Li (2020) to show that FKD offers the same generalisation benefits as found in vanilla KD in a multi-view data setting, and is further amenable to ensemble distillation. We then derive practical considerations from our insights in Section 4, to improve FKD through an understanding of the NN's feature learning optimisation process, compared to previous methods which implicitly used the feature kernel for KD. Finally, in Section 5, we provide experimental support that our theoretical claims extend to standard image classification settings, by: verifying that FKD is amenable to ensemble distillation; can transfer knowledge across datasets with different prediction spaces (unlike vanilla KD); and outperforms vanilla KD & previous feature kernel based distillation methods over a range of architectures on CIFAR-100 and ImageNet-1K.

## 2 MOTIVATION FOR FEATURE KERNEL DISTILLATION

One obvious limitation of vanilla KD is that student $f_{\mathcal{S}}$ and teacher $f_{\mathcal{T}}$ need to share prediction spaces, i.e. $C_{\mathcal{S}}=C_{\mathcal{T}}$. In many situations, we may have a teacher network trained on a dataset with a different number of classes than the student's dataset, and it is not clear how one could apply vanilla knowledge distillation. One

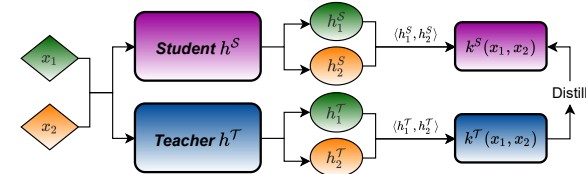

**Figure 1:** Feature Kernel Distillation (FKD) from the feature extractor of a teacher $h^{\mathcal{T}}$ to that of a student $h^{\mathcal{S}}$.

possibility could be to regularise directly in feature space by comparing $h_{\mathcal{S}}$ and $h_{\mathcal{T}}$ element-wise, but again this either requires same teacher-student last-layer sizes or additional projection layers.

To eschew such unnecessary complications, we take the perspective of NNs as data-dependent kernel machines. Define an NN's *feature kernel* to be:

**Definition 1** (Feature Kernel). *Suppose we have parameters $\boldsymbol{\theta}$ and last-layer NN feature extractor $h(\cdot, \boldsymbol{\theta}):\mathbb{R}^d \mapsto \mathbb{R}^m$. For two inputs $\boldsymbol{x}_i, \boldsymbol{x}_j \in \mathbb{R}^m$, the feature kernel $k$ is the kernel induced by the inner product of $h(\boldsymbol{x}_i, \boldsymbol{\theta})$ and $h(\boldsymbol{x}_j, \boldsymbol{\theta})$, that is: $k(\boldsymbol{x}_i, \boldsymbol{x}_j) \stackrel{\text{def}}{=} \langle h(\boldsymbol{x}_i, \boldsymbol{\theta}), h(\boldsymbol{x}_j, \boldsymbol{\theta})\rangle$.*

At initialisation, it is well known that in the infinite NN-width limit, with appropriate scaling, the feature kernel $k$ converges almost surely to a deterministic kernel known as the Neural Network Gaussian Process (NNGP) kernel (Neal, 2012; Lee et al., 2018; Matthews et al., 2018; Yang, 2019). Yang & Hu (2020) show that there is a parameterisation-dependent dichotomy between kernel & feature learning regimes for infinite-width NNs, where the feature kernel $k$ is constant or changes during training, respectively. It has been widely demonstrated that a crucial component of the success of finite-width NNs is their ability to flexibly learn features, and indeed the feature kernel, from data during training (Fort et al., 2020; Aitchison, 2020; Chen et al., 2020b; Maddox et al., 2021).

To see the importance of the feature kernel, note that for a fixed $\boldsymbol{\theta}$ with many common loss functions $L$, and some mild assumptions on strong convexity (which could be enforced e.g. with standard $\ell_2$ regularisation), the optimal $W$ is uniquely determined and $k$ determines the entire predictive function $f^*(\cdot)$. For example, with squared error, $L(f(\boldsymbol{x}), \boldsymbol{y}) = \|f(\boldsymbol{x}) - \boldsymbol{y}\|_2^2$, and $\ell_2$ regularisation strength $\lambda > 0$, a trained NN is precisely kernel ridge regression with the data-dependent feature kernel $k$, whose job is to measure how similar different inputs are. Thus, *all* teacher knowledge is contained in its feature kernel, $k^{\mathcal{T}}$, so the feature kernel can act as our primary distillation target, as depicted in Fig. 1. We show a corresponding result for cross-entropy loss in App. A.

Fig. 2 corroborates our claims. For a ResNet20v1 (He et al., 2016) 'reference' model trained on CI-FAR10 with cross entropy, we plot test class prediction confusion matrices between said model and:

i) a retrained version where all but the last-layer parameters are fixed (hence fully determined by the reference model's feature kernel as per App. A), & ii) an independent model trained from a different initialisation. As expected, there is significantly more disagreement across test predictions for models with different initialisations than those which share feature kernels. This suggests: a) different initialisations bias the same architecture to learn different

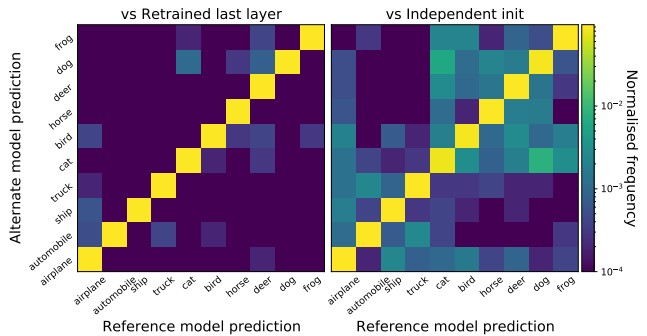

**Figure 2:** CIFAR10 test prediction confusion matrices between a fixed reference model and a model with: (left) retrained last layer, and (right) independent initialisation.

features, and b) the feature kernel (largely) determines a model's test predictions. Experimental details and a breakdown of the predictive disagreements can be found in App. F.

Having described the feature kernel as a central object in any NN, we use this to motivate our proposed FKD, where we treat the teacher's feature kernel, $k^{\mathcal{T}}$, as a key distillation target for the student's feature kernel, $k^{\mathcal{S}}$. Encouraging similarity across feature kernels shares useful features that the teacher has learnt with the student, which we theoretically show in Section 3.

We define the FKD student loss function via an additive regularisation term between feature kernels:[1]

$$\mathcal{L}^{\lambda_{\mathrm{KD}}}(\boldsymbol{\theta}, W) \stackrel{\text{def}}{=} \mathcal{L}(\boldsymbol{\theta}, W) + \lambda_{\mathrm{KD}} \cdot \mathbb{D}(k_{\boldsymbol{\theta}}, k^{\mathcal{T}}) \tag{2}$$

where $\lambda_{\mathrm{KD}} > 0$ is the regularisation strength, $\mathbb{D}$ is some (pseudo-)distance over kernels, and the student feature kernel $k^{\mathcal{S}} = k_{\boldsymbol{\theta}}$ is written to make explicit the dependence on student feature extractor parameters $\boldsymbol{\theta}$. We stress that Eq. (2) does not require matching prediction (nor feature) spaces between teacher and student, allowing us to apply FKD across tasks, architectures, and datasets.

We consider Eq. (2) with $\mathbb{D}$ set to:

$$\mathbb{D}(k_{\boldsymbol{\theta}}, k^{\mathcal{T}}) = \mathbb{E}_{\boldsymbol{x}_1, \boldsymbol{x}_2 \overset{\text{i.i.d.}}{\sim} \hat{\mathcal{D}}} \big[ \big( k_{\boldsymbol{\theta}}(\boldsymbol{x}_1, \boldsymbol{x}_2) - k^{\mathcal{T}}(\boldsymbol{x}_1, \boldsymbol{x}_2) \big)^p \big], \tag{3}$$

with expectation approximated by an average over a minibatch. In this work we choose $p = 2$, so that $\sqrt{\mathbb{D}}$ gives the Frobenius norm of the difference in feature kernel gram matrices over a batch.

## 3 THEORETICAL ANALYSIS FOR FKD

We now adapt the theoretical framework of Allen-Zhu & Li (2020), which is restricted to vanilla KD, to demonstrate the generalisation benefits of FKD over standard training. Note that FKD distils knowledge by comparing different data points, whereas vanilla KD compares a single data point across classes: this core difference is reflected throughout our analysis relative to Allen-Zhu & Li (2020). We first describe the multi-view data setting & CNN architecture we consider, before recalling that standard training without KD fails to generalise well. We then provide our main theoretical result, Theorem 2, which shows that FKD improves student test performance. Though our theoretical results are limited to a specific scenario, inspired by real-world data (Allen-Zhu & Li, 2020), & NN architecture, we believe the setup we consider is apt: it is simple enough to be tractable, yet rich enough to display the merits of FKD. Moreover, we find in Section 5 that our conclusions generalise to standard architectures & image datasets. In the interest of space & readability, we focus on providing intuition in this section, and fill in remaining details/proofs in the appendix.

---

[1]We will sometimes drop the student $\mathcal{S}$ sub/superscript where obvious for clarity, like in Eq. (2). Any teacher specific object, e.g. $k^{\mathcal{T}}$, will always have corresponding $\mathcal{T}$ sub/superscript.

**Multi-view data.** We consider the data classification problem introduced by Allen-Zhu & Li (2020), with C classes and inputs $\boldsymbol{x}$ with $P$ patches each of dimension $d$, meaning $\boldsymbol{x} \in (\mathbb{R}^d)^P$. For each class $c$, we suppose that there exist two attributes $v_{c,1}, v_{c,2} \in \mathbb{R}^d$. For $\boldsymbol{x}$ belonging to class $c$, the attributes found in patches of $\boldsymbol{x}$ will include $v_{c,1}$ and $v_{c,2}$, as well as a random selection of out-of-class attributes $\{v_{c',l}\}_{c' \neq c, l \in [2]}$.[2] This denotes the multi-view nature of the data distribution. In the true data-generating distribution $\mathcal{D}$, we suppose that a proportion $\mu$ of the data $(\boldsymbol{x}, \boldsymbol{y})$ is single-view, which means that only one of $v_{c,1}$ or $v_{c,2}$ is present in $\boldsymbol{x}$ when $(\boldsymbol{x}, \boldsymbol{y})$ is from class $c$. These will be the data for which standard training fails to generalise. A precise definition of multi-view data is presented in App. B.1.1. Allen-Zhu & Li (2020) argue that this multi-view setting provides a compelling proxy for standard image datasets such as CIFAR-10/100 Krizhevsky (2009).[3]

> **Intuition: FKD on multi-view data**
>
> Suppose we have an image classification task, with cat & car just two out of many classes. For the car class, $v_{c,1}$ could correspond to headlights, whilst $v_{c,2}$ could correspond to wheels. We would then expect $v_{c,1}$ to also appear in patches of an input image, $\boldsymbol{x}_{\text{cat}}$, corresponding to a cat with headlight-like eyes. Allen-Zhu & Li (2020) show that a single trained model is biased to learn exactly one of $v_{c,1}$ or $v_{c,2}$, depending on its parameter initialisation. W.L.O.G, suppose that the student is biased to learn $v_{c,2}$ & not $v_{c,1}$. If the teacher model has learnt $v_{c,1}$, this means that the teacher model knows there is a similarity between $\boldsymbol{x}_{\text{cat}}$ & any car image, $\boldsymbol{x}_{\text{car}}$, that displays headlights. Mathematically, we show that this corresponds to a large value for $k_{\mathcal{T}}(\boldsymbol{x}_{\text{cat}}, \boldsymbol{x}_{\text{car}})$. Our FKD regularisation forces the student to also have a large value for $k_{\mathcal{S}}(\boldsymbol{x}_{\text{cat}}, \boldsymbol{x}_{\text{car}})$, ensuring that attribute $v_{c,1}$ is also learnt by the student network. Without distillation, a student NN which has learnt $v_{c,2}$ & not $v_{c,1}$ will not generalise to front-on images of cars that hide wheels.

**Convolutional NN & corresponding feature kernel.** Like Allen-Zhu & Li (2020), for our theoretical analyis we consider a single hidden-layer convolutional NN (CNN) with sum-pooling.[4] For each class $c \in [C]$, we suppose that the CNN has $m$ channels, giving $Cm$ channels in total. For channel $r$ and class $c$, we suppose that we have weights $\boldsymbol{\theta}_{c,r} \in \mathbb{R}^d$. This gives output for class $c$ by

$$f_c(\boldsymbol{x}) \stackrel{\text{def}}{=} \sum_{r=1}^{m} \sum_{p=1}^{P} \widetilde{\text{ReLU}}(\langle \boldsymbol{\theta}_{c,r}, \boldsymbol{x}_p \rangle) \tag{4}$$

where for ease of analysis $\widetilde{\text{ReLU}}$ is ReLU-like but with continuous gradient, see App. B.2.

Before we consider FKD, we must first define the feature kernel for this CNN $f$. To do so, we recast $f(\boldsymbol{x}) = W \cdot h(\boldsymbol{x}, \boldsymbol{\theta})$, where $h(\boldsymbol{x}, \boldsymbol{\theta}) \in \mathbb{R}^{Cm}$ and $W \in \mathbb{R}^{C \times Cm}$ satisfying, for $r \in [m], c \in [C], c' \in [C]$:

$$h(\boldsymbol{x}, \boldsymbol{\theta})_{r+(c-1)m} \stackrel{\text{def}}{=} \sum_{p=1}^{P} \widetilde{\text{ReLU}}(\langle \boldsymbol{\theta}_{c,r}, \boldsymbol{x}_p \rangle), \quad \text{and} \quad W_{c',r+(c-1)m} \stackrel{\text{def}}{=} \mathbb{1}\{c = c'\}. \tag{5}$$

Given that the feature kernel is $k(\boldsymbol{x}, \boldsymbol{x}') \stackrel{\text{def}}{=} \langle h(\boldsymbol{x}, \boldsymbol{\theta}), h(\boldsymbol{x}', \boldsymbol{\theta}) \rangle$, we now have that:[5]

$$k(\boldsymbol{x}, \boldsymbol{x}') = \sum_{c=1}^{C} \sum_{r=1}^{m} \sum_{p,p'=1}^{P} \widetilde{\text{ReLU}}(\langle \boldsymbol{\theta}_{c,r}, \boldsymbol{x}_p \rangle) \cdot \widetilde{\text{ReLU}}(\langle \boldsymbol{\theta}_{c,r}, \boldsymbol{x}'_{p'} \rangle). \tag{6}$$

We first recall that standard training of the model $f$ with gradient descent and cross entropy loss fails to generalise on half the $\mu$ proportion of data that is single-view.

---

[2]It is straightforward to extend to the case of more than two views per class if need be.

[3]https://www.microsoft.com/en-us/research/blog/three-mysteries-in-deep-learning-ensemble-knowledge-distillation-and-self-distillation/

[4]It is straightforward to extend our analysis for max-pooling.

[5]The feature kernel defined in Eq. (6) corresponds to the Global Average Pooling CNN-GP kernel in Novak et al. (2018) in the infinite-channel limit, which captures intra-patch correlations unlike the vectorised CNN-GP, which corresponds to vectorising the spatial dimensions to give $CmP$ rather than $Cm$ channels.

**Theorem 1** (Standard training fails, Theorem 1 of Allen-Zhu & Li (2020)). *For sufficiently many classes $C$ and channels $m \in [\mathrm{polylog}(C), C]$, with learning rate $\eta \leq \frac{1}{\mathrm{poly}(C)}$, training time $T^* = \frac{\mathrm{poly}(C)}{\eta}$, and multi-view data distribution (App. B.1.1), the trained model $f^{(T^*)}$ satisfies with probability at least $1 - e^{-\Omega(\log^2(C))}$:*

- *Training accuracy is perfect: For all $(\boldsymbol{x}, \boldsymbol{y}) \in \hat{\mathcal{D}}$, $\boldsymbol{y} = \mathrm{argmax}_c f_c^{(T^*)}(\boldsymbol{x})$.*

- *Test accuracy is bad but consistent: $\mathbb{P}_{(\boldsymbol{x}, \boldsymbol{y}) \sim \mathcal{D}} \big[ \boldsymbol{y} \neq \mathrm{argmax}_c f_c^{(T^*)}(\boldsymbol{x}) \big] \in [0.49\mu, 0.51\mu]$.*

Now we are ready to show that regularising the student model towards the teacher model's feature kernel, as in FKD, improves test accuracy. We suppose our teacher model is an ensemble of $E \geq 1$ models $\{f_e\}_{e=1}^E$, each with corresponding feature kernel $k_e$, trained as standard on the same data with independent initialisations $\boldsymbol{\theta}_0^e$. We average $k_e$ over $e \in [E]$ to obtain our teacher feature kernel:

$$k_{\mathcal{T}}(\boldsymbol{x}, \boldsymbol{x}') = \frac{1}{E} \sum_{e=1}^E k_e(\boldsymbol{x}, \boldsymbol{x}'). \tag{7}$$

This is akin to concatenating all features in $\{h_e\}_{e=1}^E$ into a $ECm$-dimensional feature vector, albeit without the additional computational baggage. We then have our main theoretical result:

**Theorem 2** (FKD improves student generalisation and is better with larger ensemble). *Given an arbitrary $\epsilon > 0$. For any ensemble size $E$ of teacher NNs trained as in Theorem 1 and sufficiently many classes $C$, for $m = \mathrm{polylog}(C)$, with learning rate $\eta \leq \frac{1}{\mathrm{poly}(C)}$, and training time $T^* = \frac{\mathrm{poly}(C)}{\eta}$, the ensemble teacher knowledge can be distilled into a single student model $f^{(T^*)}$ using only teacher feature kernel $k_{\mathcal{T}}$, Eq. (7), such that with probability at least $1 - e^{-\Omega(\log^2(C))}$:*

- *Training accuracy is perfect: For all $(\boldsymbol{x}, \boldsymbol{y}) \in \hat{\mathcal{D}}$, $\boldsymbol{y} = \mathrm{argmax}_c f_c^{(T^*)}(\boldsymbol{x})$.*

- *Test accuracy is good: $\mathbb{P}_{(\boldsymbol{x}, \boldsymbol{y}) \sim \mathcal{D}} \big[ \boldsymbol{y} \neq \mathrm{argmax}_c f_c^{(T^*)}(\boldsymbol{x}) \big] \leq (\frac{1}{2^{E+1}} + \epsilon)\mu$.*

*Proof outline for Theorem 2*, we first show, in Lemma 1, that a single trained NN's feature kernel (which we defined in Eq. (6)) can detect if two inputs share a data attribute that the NN has learnt due to its weight initialisation. We extend this result to an ensemble teacher in App. C.2, showing that the ensemble teacher feature kernel detects the union of all data attributes learnt by individual $\{k_e\}_{e=1}^E$. This simplifies our calculations when showing that our distillation regulariser, Eq. (3), is effective for improved student generalisation. The full proof can be found in App. C.

---

**Intuition: FKD with ensemble of teachers**

To parse Theorem 2, suppose we have a single teacher i.e. $E=1$. Then, the test error is essentially $0.25\mu$ in Theorem 2. The explanation is that both the student & teacher networks independently learn one of $\{v_{c,l}\}_{l=1}^2$ for each class $c$. Either vanilla KD (Theorem 4 of Allen-Zhu & Li (2020)) or our feature kernel approach allow the student to learn the union of the independent attributes learnt by the student and the teacher, so for only a quarter of the single-view test data $\boldsymbol{x}_s$ will the student not have learnt the useful class attribute present in $\boldsymbol{x}_s$. For general ensemble size $E$, the story is the same: the student & $E$ teachers each independently learn one of the two useful attributes $\{v_{c,l}\}_{l \in 2}$ for all $c \in [C]$. Distilling allows the student to learn the union of these attributes, which means that the student will fail on only $\frac{1}{2^{E+1}}$ of the single-view data.

---

## 4 FKD IN PRACTICE

Next, we highlight practical considerations for implementing FKD derived from our theory. Pseudo-code and PyTorch-style code for our FKD implementation are given in Algs. 1 and 2 respectively.

**Correlation kernel.** We propose $\mathbb{D}(\rho_{\boldsymbol{\theta}}, \rho_{\mathcal{T}})$ as a regulariser in FKD instead of $\mathbb{D}(k_{\boldsymbol{\theta}}, k_{\mathcal{T}})$, where:

$$\rho_z(\boldsymbol{x}, \boldsymbol{x}') \stackrel{\text{def}}{=} \frac{k_z(\boldsymbol{x}, \boldsymbol{x}')}{\sqrt{k_z(\boldsymbol{x}, \boldsymbol{x})k_z(\boldsymbol{x}', \boldsymbol{x}')}}, \quad \text{and} \quad \rho_{\mathcal{T}}(\boldsymbol{x}, \boldsymbol{x}') \stackrel{\text{def}}{=} \frac{1}{E}\sum_{e=1}^{E} \rho_e(\boldsymbol{x}, \boldsymbol{x}')$$

defines *feature correlation kernel* $\rho_z$, corresponding to feature kernel $k_z$, for $z \in [E] \cup \{\boldsymbol{\theta}\}$. The reason we use correlation kernels is that they normalise data, so that $\rho(\boldsymbol{x}, \boldsymbol{x}) = 1 \ \forall \boldsymbol{x}$, which zeros diagonal differences in Eq. (3): we hope FKD allows the student to learn from the teacher features shared between *different* inputs. Non-zero diagonal differences, like in Similarity-Preserving (SP) KD (Tung & Mori, 2019), encourage the student to learn noise as we show in App. D.1, and we hypothesise that this contributes to the improved performance we observe of FKD over SP in Section 5. Moreover, this normalisation helps balance individual teacher's influence in an ensemble teacher, and ensures that FKD does not need a temperature hyperparameter $\tau$, which produces soft labels in vanilla KD.

**Feature regularisation.** One downside to using the correlation kernel is that our FKD regularisation, $\mathbb{D}(\rho_{\boldsymbol{\theta}}, \rho_{\mathcal{T}})$, becomes invariant to the scale of $k_{\boldsymbol{\theta}}$. For example, replacing $k_{\boldsymbol{\theta}}(\boldsymbol{x}, \boldsymbol{x}')$ with $\sqrt{M(\boldsymbol{x})M(\boldsymbol{x}')}k_{\boldsymbol{\theta}}(\boldsymbol{x}, \boldsymbol{x}')$, for any $M(\boldsymbol{x}):\mathbb{R}^d \mapsto \mathbb{R}^+$, leaves the student correlation kernel unchanged. This may lead to degeneracies when training $\boldsymbol{\theta}$, & large variations in $k(\boldsymbol{x}, \boldsymbol{x})$ over $\boldsymbol{x}$ may harm generalisation (as evidenced by the fact that input normalisation is common across ML, from linear models to NNs). Moreover, our proof of Theorem 2 is not quantitative, in that we only show $k_{\boldsymbol{\theta}}(\boldsymbol{x}, \boldsymbol{x}') = \tilde{\Theta}(1)$ in the number of classes $C$ *up to* polylogarithmic factor in $C$, when $\boldsymbol{x} \neq \boldsymbol{x}'$ share a data attribute $v_{c,l}$ that has been learned by parameters $\boldsymbol{\theta}$. These insights motivate us to regularise the student feature $h$ during FKD

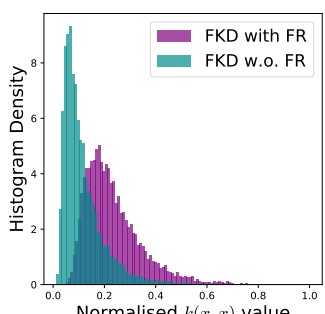

**Figure 3:** Histogram of normalised feature kernel values, $\frac{k(\boldsymbol{x}, \boldsymbol{x})}{\max_{\boldsymbol{x}'} k(\boldsymbol{x}', \boldsymbol{x}')}$, over the CIFAR-100 test set.

training to control the norm of $k(\boldsymbol{x}, \boldsymbol{x})$ across inputs $\boldsymbol{x}$. We use an additive $\ell_2$ regularisation $\frac{1}{B}\sum_{b=1}^{B} \|h(\boldsymbol{x}_b, \boldsymbol{\theta})\|_2^2 = \frac{1}{B}\sum_{b=1}^{B} k(\boldsymbol{x}_b, \boldsymbol{x}_b)$ with regularisation strength $\lambda_{\text{FR}} > 0$, for each minibatch $\{(\boldsymbol{x}_b, \boldsymbol{y}_b)\}_{b=1}^{B}$. Fig. 3 shows that using Feature Regularisation (FR) encourages a more even spread of $k(\boldsymbol{x}, \boldsymbol{x})$ across inputs for a student VGG8 network trained with a VGG13 teacher model on CIFAR-100. Corresponding plots for other architectures can be found in Fig. 7. Similar to Dauphin & Cubuk (2021), we find that FR improves generalisation & provide ablations in Section 5.

---

**Algorithm 1** Feature Kernel Distillation with SGD.

---

**Require:** Maximum number of iterations $T^*$, batch size $B$, learning rate $\eta$, teacher correlation kernel $\rho_{\mathcal{T}}$, FKD regularisation $\lambda_{\text{KD}} > 0$, Feature regularisation $\lambda_{\text{FR}} > 0$. Initialise student parameters $\boldsymbol{\theta}_0, W_0$.
  **for** iteration $t = 0, \ldots, T^*$ **do**
    Sample minibatch $(\boldsymbol{x}_i^B, y_i^B)_{i=1}^{B} \stackrel{\text{i.i.d.}}{\sim} \hat{\mathcal{D}}$.
    Compute loss $\mathcal{L} = \frac{1}{B}\sum_{i=1}^{B} L(f(\boldsymbol{x}_i^B), y_i^B) + \frac{2\lambda_{\text{KD}}}{B(B-1)}\sum_{i \neq j}^{B} \left(\rho_{\boldsymbol{\theta}_t}(\boldsymbol{x}_i^B, \boldsymbol{x}_j^B) - \rho_{\mathcal{T}}(\boldsymbol{x}_i^B, \boldsymbol{x}_j^B)\right)^2$.
    Add feature regularisation $\mathcal{L} = \mathcal{L} + \frac{\lambda_{\text{FR}}}{B}\sum_{i=1}^{B} \|h(\boldsymbol{x}_i^B, \boldsymbol{\theta}_t)\|_2^2$, (optional).
    Update parameters $\boldsymbol{\theta}_{t+1} \leftarrow \boldsymbol{\theta}_t - \eta\nabla_{\boldsymbol{\theta}}\mathcal{L}$, $W_{t+1} \leftarrow W_t - \eta\nabla_W\mathcal{L}$.
  **end for**
  **return** $\{\boldsymbol{\theta}_{T^*}, W_{T^*}\}$

---

## 5 EXPERIMENTS

Due to space concerns, App. F contains further experiments & any missing experimental details.

**Ensemble distillation.** We first verify that larger ensemble teacher size, $E$, further improves FKD student performance as suggested by Theorem 2. This is confirmed in Fig. 4, using VGG8 for all student & teacher networks on the CIFAR-100 dataset.

We also plot the test accuracy of the ensemble teacher across sizes $E$, whose predictive probabilities are averaged over individual teachers, as well as the test accuracy of an undistilled student model. We see that FKD consistently outperforms vanilla KD, and both distillation methods outperform the teacher in the 'self-distillation' setting of $E=1$ (Furlanello et al., 2018; Zhang et al., 2019). Moreover, FKD allows a single student to match teacher performance when $E=2$, before positive but diminishing returns with larger $E$ relative to the teacher ensemble.

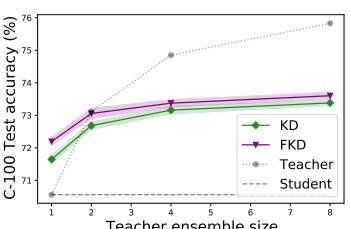

**Figure 4:** FKD as teacher ensemble size changes. Error bars denote 95% confidence for mean of 10 runs.

**Dataset Transfer.** We next show FKD can transfer knowledge across similar datasets. From a fixed VGG13 teacher network trained on CIFAR-100, we distil to student VGG8 NNs on CIFAR-10, STL-10 & Tiny-ImageNet.

**Table 1:** Test accuracy (%) of FKD & baselines in a dataset transfer distillation setting. Error bars indicate 95% confidence for mean of 10 runs.

| Dataset | Student | RKD | SP | FKD w.o. FR | FKD |
|---|---|---|---|---|---|
| **C-100 → C-10** | 91.56 | $91.74_{\pm0.09}$ | $92.21_{\pm0.07}$ | $92.33_{\pm0.07}$ | $\mathbf{92.61_{\pm0.07}}$ |
| **C-100 → STL-10** | 72.69 | $72.86_{\pm0.27}$ | $73.88_{\pm0.17}$ | $\mathbf{75.17_{\pm0.30}}$ | $\mathbf{75.44_{\pm0.31}}$ |
| **C-100 → Tiny-I** | 48.53 | $48.74_{\pm0.17}$ | $48.73_{\pm0.14}$ | $48.85_{\pm0.09}$ | $\mathbf{50.67_{\pm0.12}}$ |

As no student dataset has 100 classes, unlike CIFAR-100, it is not clear how one can use vanilla KD (Hinton et al., 2015) in this case. We thus compare FKD to other feature kernel based KD methods: Relational KD (RKD) (Park et al., 2019) & Similarity-Preserving (SP) KD (Tung & Mori, 2019). In Table 1, we see that FKD without feature regularisation outperforms both baselines across all datasets, and that feature regularisation (FR) further improves FKD performance, highlighting the benefit of our practical considerations in Section 4. The improved performance is particularly stark on STL-10 (which we downsize to 32x32 resolution), where FKD improves student performance by 2.75%. STL-10 is well suited for FKD as it has only 5K labeled inputs but 100K unlabeled datapoints, which can be used in our feature kernel regulariser, $\mathbb{D}(\rho_{\boldsymbol{\theta}}, \rho_{\mathcal{T}})$.

**Table 2:** CIFAR-100 and ImageNet-1K accuracies (%) comparing FKD with KD baselines. * denotes result from Tian et al. (2020); FKD uses the same teacher checkpoints provided by the authors,[6] with error bars denoting 95% confidence for the mean over 5 students.

| | **CIFAR-100** | | | | **ImageNet-1K** |
|---|---|---|---|---|---|
| **Teacher** | ResNet32x4 | VGG13 | ResNet32x4 | ResNet50 | ResNet34 |
| **Student** | ResNet8x4 | VGG8 | ShuffleNetV1 | VGG8 | ResNet18 |
| **Teacher*** | 79.42 | 74.64 | 79.42 | 79.34 | 73.26 |
| **Student*** | 72.50 | 70.36 | 70.50 | 70.36 | 69.97 |
| **KD* (Hinton et al., 2015)** | $73.33_{\pm0.22}$ | $72.98_{\pm0.17}$ | $74.07_{\pm0.17}$ | $73.81_{\pm0.11}$ | 70.66 |
| **RKD* (Park et al., 2019)** | $71.90_{\pm0.10}$ | $71.48_{\pm0.04}$ | $72.28_{\pm0.34}$ | $71.50_{\pm0.06}$ | N/A |
| **SP* (Tung & Mori, 2019)** | $72.94_{\pm0.20}$ | $72.68_{\pm0.17}$ | $73.48_{\pm0.37}$ | $73.34_{\pm0.30}$ | 70.62 |
| **CRD* (Tian et al., 2020)** | $\mathbf{75.51_{\pm0.16}}$ | $\mathbf{73.94_{\pm0.19}}$ | $\mathbf{75.11_{\pm0.28}}$ | $74.30_{\pm0.12}$ | 71.17 |
| **FKD w.o. FR** | $74.89_{\pm0.24}$ | $73.08_{\pm0.16}$ | $74.66_{\pm0.23}$ | $73.99_{\pm0.15}$ | 70.84 |
| **FKD** | $\mathbf{75.57_{\pm0.22}}$ | $\mathbf{73.78_{\pm0.17}}$ | $\mathbf{75.00_{\pm0.30}}$ | $\mathbf{74.61_{\pm0.28}}$ | $\mathbf{71.23}$ |

**Comparison on CIFAR-100 and ImageNet.** Finally, we compare FKD to various knowledge distillation baselines on CIFAR-100 and ImageNet, across a selection of teacher/student architectures. We see in Table 2 that FKD consistently outperforms: vanilla KD (Hinton et al., 2015), RKD (Park et al., 2019), and SP (Tung & Mori, 2019). Moreover, FKD either matches or outperforms the high-performing Contrastive Representational Distillation (Tian et al., 2020). We use the exact same teacher checkpoints used by Tian et al. (2020) and Chen et al. (2021) for CIFAR-100 and ImageNet respectively to ensure fair comparison. We find, like in Table 1, that feature regularisation consistently improves FKD performance and that even without feature regularisation, FKD outperforms all feature kernel based KD methods. This implies that using the correlation kernel to zero out diagonal differences, as described in Section 4, indeed helps improve student performance.

---

[6]Apart from ImageNet-1K which used the pretrained ResNet34 from torchvision, like Chen et al. (2021)

## 6   RELATED WORK

**NN Knowledge Distillation.**    Following Hinton et al. (2015), there has been much interest in expanding KD in NNs (Romero et al., 2014; Zagoruyko & Komodakis, 2016; Passalis & Tefas, 2018; Zhang et al., 2018; Yu et al., 2019; Chen et al., 2020a; Tian et al., 2020). Most similar to FKD are Park et al. (2019); Tung & Mori (2019) who also use relations between inputs to distil knowledge (albeit not from the feature kernel learning perspective and without our theoretical justification), as well as Qian et al. (2020) who focus on reducing computational costs of full-batch kernel matrix operations. App. D highlights in more detail the differences of FKD compared to previous pairwise feature kernel based KD methods. Allen-Zhu & Li (2020) made the first theoretical connection using the mechanisms of ensembling in NNs to explain the success of vanilla KD in NNs, which we extend for feature kernel based KD.

**Ensembling NNs.**    Ensembling NNs has long been studied for improving predictive accuracy (Hansen & Salamon, 1990; Krogh et al., 1995) with particular recent focus towards uncertainty quantification & Bayesian inference (Lakshminarayanan et al., 2017; Ovadia et al., 2019; Zaidi et al., 2020; Pearce et al., 2020; He et al., 2020; Wilson & Izmailov, 2020; Wenzel et al., 2020; D'Angelo & Fortuin, 2021; Schut et al., 2021) and predictive diversity (Fort et al., 2019; D'Amour et al., 2020). On the topic of Bayesian inference, the feature kernel has also been studied under the name of Neural Linear Model (Riquelme et al., 2018; Ober & Rasmussen, 2019), and extensions treating features $h$ as inputs to standard Gaussian Process kernels are known by the name of Deep Kernel Learning (Wilson et al., 2016; Ober et al., 2021; van Amersfoort et al., 2021).

**NN Feature learning.**    A recent flurry of work has focussed on characterising & understanding the importance of feature learning in NNs (Chizat et al., 2019; Fort et al., 2020; Baratin et al., 2021; Lee et al., 2020; Aitchison, 2020; Ghorbani et al., 2020), fuelled in part by the development that wide NNs become (Neural Tangent) Kernel machines in certain regimes (Jacot et al., 2018; Lee et al., 2019; Yang & Littwin, 2021), thus forgoing feature learning. The consensus in these works is that there are gaps between NTK theory & practical NNs that cannot be explained without feature learning. However, Yang & Hu (2020) proved that feature-learning is still possible with infinite-width NNs, and also that feature learning is equivalent to feature kernel learning in infinite-width NNs. This motivates our study of the feature kernel as a key object for distillation. Regularising the feature kernel to the true target covariance kernel was suggested by Yoo et al. (2021).

## 7   CONCLUSION

We have theoretically shown that the feature kernel is a valid object for Knowledge Distillation (KD) in Neural Networks (NNs) by extending the analysis of Allen-Zhu & Li (2020), which focused on vanilla KD (Hinton et al., 2015). Further, we used our theoretical insights to motivate practical considerations when using feature kernels for distillation, such as using the feature correlation kernel & using feature regularisation, to improve on previous feature based KD methods. We term our approach Feature Kernel Distillation (FKD), and note that FKD is more widely applicable than vanilla KD, as it benefits from being agnostic to teacher and student prediction spaces. Experimentally, we have demonstrated that FKD is amenable to ensemble distillation as suggested by our theory, is able to transfer knowledge across similar datasets and that FKD outperforms vanilla KD & previous feature kernel based KD methods across a variety of architectures on CIFAR-100, and ImageNet-1K.

**Limitations & future work.**    Though feature learning is central to our results, we stress that there are still gaps between our theory & practice to understanding NN ensembling & KD, demonstrated by the divergence between ensemble teacher & FKD in Fig. 4 for larger ensemble size. This could be due to: the multi-view data setting not being able to capture the full complexity of real-world data, the role of hierarchical feature learning between layers in a deep NN, or the importance of mini-batching in stochastic gradient descent. Other future work could apply the multi-view data setting to analyse uncertainty quantification in NN ensembles, assess the impact of different FKD regularisation metrics in Eq. (3), or improve FKD further to compete with state-of-the-art KD methods.

**Acknowledgements**  We thank Emilien Dupont, Yee Whye Teh, and Sheheryar Zaidi for helpful feedback on this work.

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

# APPENDIX: FEATURE KERNEL DISTILLATION

## A  FEATURE KERNEL DEPENDENCE WITH CROSS-ENTROPY LOSS

Suppose, we have $n$ data points with fixed feature extractor $h = h(\boldsymbol{X}) \in \mathbb{R}^{n \times p}$, trainable last layer weights $W \in \mathbb{R}^{p \times C}$ and targets $Y \in \mathbb{R}^{n \times C}$.

In Section 2, we described how with Mean Squared Error loss the trained NN is precisely kernel ridge regression using the feature kernel, when there is $\ell_2$ regularisation on the last layer weights. Hence, the feature kernel exactly determines the predictions. We now prove the same for cross entropy loss. That is to say, we wish to show that given the feature kernel $k$, the predictive probabilities/logits at the optimal $W^*$ are independent of both $W^*$ and $h$.

We define the loss, for some regularisation $\lambda > 0$ (purely to enforce strong convexity) by

$$\mathcal{L}(W) = \sum_{i=1}^{n} CE(y_i, h_i W) + \frac{\lambda}{2} \|W\|_2^2,$$

where $h_i \in \mathbb{R}^{1 \times p}$ is the $i^{\text{th}}$ row of $h$ and

$$CE(y, f) = -\log \frac{e^{f_y}}{\sum_{c \in [C]} e^{f_c}}, \tag{8}$$

is cross entropy loss.

**Proposition 1.** *Let feature extractor $h(\cdot)$, training inputs $\boldsymbol{X}$, training targets $Y$, $\ell_2$ regularisation $\lambda > 0$ all be fixed. Suppose we are given a test point $\boldsymbol{x}'$ with features $h'$, then the test prediction logits $h'W^*$ at optimal $W^*$ can be expressed solely in terms of feature kernel evaluations $k(\cdot, \cdot) = \langle h(\cdot), h(\cdot) \rangle$.*

*Proof.* If we differentiate $\mathcal{L}(W)$ and set to zero we get:

$$\lambda W_{j,l}^* = \sum_{i=1}^{n} h_{i,j} \big[ \mathbb{1}\{y_i = l\} - p_{i,l} \big] \tag{9}$$

where $p \in \mathbb{R}^{n \times C}$ is the result of applying softmax to each row of the logits $h'W^*$.

Let $h_i$ denote the extracted features for training input $\boldsymbol{x}_i$. Now recall that $\langle h_i, h_j \rangle = k(\boldsymbol{x}_i, \boldsymbol{x}_j)$ is the feature kernel evaluated at $\boldsymbol{x}_i, \boldsymbol{x}_j$. We multiply Eq. (9) by the test-data feature vector $h'$ to give:

$$\lambda (h'W^*)_l = \sum_{i=1}^{n} k(\boldsymbol{x}_*, \boldsymbol{x}_i) \big[ \mathbb{1}\{y_i = l\} - p_{i,l} \big] \tag{10}$$

but $h'W^*$ are precisely the logits for test point $\boldsymbol{x}_*$, and likewise $p_{i,l}$ is the vector of probability predictions at training point $i$. Hence, we could solve Eq. (10) numerically for logits/predictive probabilities given only the feature kernel (without $h(\cdot)$ or $W^*$), and we see that the feature kernel once again determines logit/prediction probabilities at the optimal last layer parameters like for squared error loss, albeit this time implicitly for cross entropy loss.  □

# B    SETUP FOR TRAINING ON MULTI-VIEW DATA

Before we prove Theorem 2, which demonstrates the generalisation benefits of FKD theoretically, we need to recall the data and training setup of Allen-Zhu & Li (2020) for completeness and self-containedness. Unless otherwise stated, everything in this section (App. B) is a simplified version of the setup in Allen-Zhu & Li (2020). Our results hold in the more general version too, but we present the setup in a simplified setting here for readability and convenience, without sacrificing the key messages and intuitions: our focus is for our theoretical and practical contributions to guide each other and align as much as possible, as opposed to e.g. maximising the generality of our theory.

## B.1    MULTI-VIEW DATA DISTRIBUTION

Recall that we consider a $C$-class classification problem over $P$-patch inputs, where each patch has dimension $d$, so our inputs are described by $\boldsymbol{x} = (\boldsymbol{x}^1, \ldots, \boldsymbol{x}^P) \in (\mathbb{R}^d)^P$. For simplicity, we take $P = C^2$ and $d = \text{poly}(C)$ for a large polynomial. Like Allen-Zhu & Li (2020), we use $\tilde{O}, \tilde{\Theta}, \tilde{\Omega}$ to hide polylogarithmic factors in the number of classes, $C$, which we take to be sufficiently large.

We assume that each class $c \in [C]$ has exactly two attributes $v_{c,1}, v_{c,2} \in \mathbb{R}^d$, which are orthonormal for simplicity,[7] such that:

$$\mathcal{V} = \{v_{c,1}, v_{c,2}\}_{c \in [C]}$$

is the set of all attributes.

### B.1.1    DATA GENERATING MECHANISM

Let our data distribution for a data pair $(\boldsymbol{x}, \boldsymbol{y}) \sim \mathcal{D}$ be defined as $\mathcal{D} = \mu \mathcal{D}_s + (1 - \mu)\mathcal{D}_m$, for multi-view & single-view distributions $\mathcal{D}_m$ & $\mathcal{D}_s$ respectively. $(\boldsymbol{x}, \boldsymbol{y}) \sim \mathcal{D}$ are generated as follows:

1. Sample $\boldsymbol{y} \in [C]$ uniformly at random.
2. Sample a set $\mathcal{V}'(\boldsymbol{x})$ of attributes uniformly at random from $\{v_{c',1}, v_{c',2}\}_{c' \neq y}$ each with probability $\frac{s}{C}$ and denote $\mathcal{V}(X) = \mathcal{V}'(\boldsymbol{x}) \cup \{v_{y,1}, v_{y,2}\}$ as the set of attribute vectors used in data $\boldsymbol{x}$. We take $s = C^{0.2}$.

> **Intuition:**
>
> $\mathcal{V}'(\boldsymbol{x})$ correspond to the ambiguous attributes present in $\boldsymbol{x}$, such as the cat whose eyes look like car headlights. These $\mathcal{V}'$ are crucial in our proofs as the FKD regularisation between ambiguous images ensures that the student learns attributes it would have otherwise missed.

   For each $v \in \mathcal{V}(\boldsymbol{x})$, pick $C_p$ (where $C_p$ is a global constant) many disjoint patches in $[P]$ uniformly at random and denote this set as $\mathcal{P}_v(\boldsymbol{x})$. Denote $\mathcal{P}(\boldsymbol{x}) = \cup_{v \in \mathcal{V}(\boldsymbol{x})} \mathcal{P}_v(\boldsymbol{x})$: all other patches $p \notin \mathcal{P}(\boldsymbol{x})$ will contain noise only.

4. If $\boldsymbol{x}$ is **single** view, pick a value $\hat{l} = \hat{l}(\boldsymbol{x}) \in \{1, 2\}$ uniformly at random. $\hat{l}$ corresponds to the attribute $v_{y,\hat{l}}$ that is present in $\boldsymbol{x}$, with $v_{y,3-\hat{l}}$ missing.
5. For each $p \in \mathcal{P}_v(\boldsymbol{x})$ for some $v \in \mathcal{V}(\boldsymbol{x})$:

$$\boldsymbol{x}_p = z_p v + \sum_{v' \in \mathcal{V}} \alpha_{p,v'} v' + \xi_p$$

---

[7]As $d = \text{poly}(C)$ for a large polynomial, this isn't too far-fetched an assumption.

where $\alpha_{p,v'} \in [0, \frac{1}{C^{1.5}}]$ represents feature noise and $\xi_p \sim \mathcal{N}(0, \sigma_p^2 \mathbf{I}_d)$ is independent random noise, with $\sigma_p = \frac{1}{\sqrt{d}\text{polylog}(C)}$. The coefficients $z_p \geq 0$ satisfy:

(a) If $x$ is **multi** view,

- When $v \in \{v_{y,1}, v_{y,2}\}$,

$$\begin{cases} \sum_{p \in \mathcal{P}_v(x)} z_p \in [1, 2) \\ \sum_{p \in \mathcal{P}_v(x)} z_p^4 = 1 \end{cases} \tag{11}$$

> **Intuition:**
> These conditions ensure that both attributes for class $y$ are equally likely to be learnt, averaged over different random initialisations of parameters.

- When $v \in \mathcal{V}'(x)$,

$$\begin{cases} \sum_{p \in \mathcal{P}_v(x)} z_p = 0.4 \\ \sum_{p \in \mathcal{P}_v(x)} z_p^4 = \Theta(1) \end{cases} \tag{12}$$

(b) If $x$ is **single** view,

- When $v = v_{y,\hat{i}}$,     $\sum_{p \in \mathcal{P}_v(x)} z_p = 1$
- When $v = v_{y,3-\hat{i}}$,     $\sum_{p \in \mathcal{P}_v(x)} z_p = C^{-0.2}$
- When $v \in \mathcal{V}'(x)$,     $\sum_{p \in \mathcal{P}_v(x)} z_p = \Gamma$

where $\Gamma = \frac{1}{\text{polylog}(C)}$

> **Intuition:**
> This is where the single view name comes from, as $C^{-0.2} \ll 1$ we see that $v_{y,3-\hat{i}}$ is barely present in $x$.

6. For each $p \in [P] \backslash \mathcal{P}_v(x)$:

$$x_p = \sum_{v' \in \mathcal{V}} \alpha_{p,v'} v' + \xi_p$$

for feature noise $\alpha_{p,v'} \in [0, \frac{1}{C^{1.5}}]$ and $\xi_p \sim \mathcal{N}(0, \frac{1}{Cd}\mathbf{I}_d)$ is independent random noise.

> **Intuition:**
> One can think of the zero feature noise setting $\alpha_{p,v'} = 0 \ \forall p, v'$ for simplicity. But the general formulation above renders the problem unlearnable by linear classifier, as the maximum permissible feature noise across patches dominates the minimum possible signal: $\frac{P}{C^{1.5}} = C^{0.5} \gg 1$

**Remark**    It is possible to allow more relaxed assumptions, e.g. on $z_p$, as in Allen-Zhu & Li (2020).

**Training data**    Recall we have $\mathcal{D} = \mu \mathcal{D}_s + (1-\mu)\mathcal{D}_m$, so that a proportion $(1-\mu)$ of the data is multi-view. Our training data, $\hat{\mathcal{D}}$, is $N$ independent samples from $\mathcal{D}$. Letting $\hat{\mathcal{D}} = \hat{\mathcal{D}}_m \cup \hat{\mathcal{D}}_s$ denote the split into multi and single view training data. We let $\mu = \frac{1}{\text{poly}(C)}$ and we suppose $|N| = \frac{C^{1.2}}{\mu}$ so that each label $c$ appears at least $\Omega(1)$ in $\hat{\mathcal{D}}_s$.

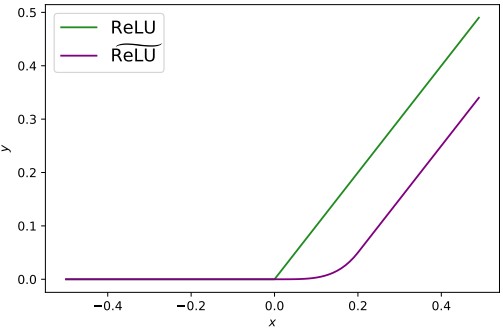

**Figure 5:** Comparison between ReLU & $\widetilde{\mathrm{ReLU}}$, for $\varrho = 0.2$.

## B.2 SMOOTHED RELU CNN

Our theoretical analysis considers a single hidden layer CNN with sum-pooling, $f$, such that for class $c$:

$$f_c(\boldsymbol{x}) = \sum_{r=1}^{m} \sum_{p=1}^{P} \widetilde{\mathrm{ReLU}}(\langle \boldsymbol{\theta}_{c,r}, \boldsymbol{x}_p \rangle), \quad \forall c \in [C].$$

For a threshold $\varrho = \frac{1}{\mathrm{polylog}(C)}$, we define the smoothed ReLU function as:

$$\widetilde{\mathrm{ReLU}}(z) = \begin{cases} 0 & \text{if } z \leq 0 \\ \frac{z^4}{4\varrho^3} & \text{if } z \in [0, \varrho] \\ z - \frac{3}{4}\varrho & \text{if } z \geq \varrho \end{cases}$$

which has continuous monotonic gradient denoted as $\widetilde{\mathrm{ReLU}}'$.

## B.3 HYPERPARAMETER VALUES IN SETUP

Hyperparameter values used in the theoretical setup are given in Table 3.

## B.4 STANDARD TRAINING

We define our empirical loss by:

$$\mathcal{L} = \frac{1}{N} \sum_{i \in [N]} L(f(\boldsymbol{x}_i), \boldsymbol{y}_i)$$

where $L$ is the cross entropy loss defined in Eq. (8). We randomly initialise parameters $\boldsymbol{\theta}_{c,r}^0 \overset{\text{i.i.d.}}{\sim} \mathcal{N}(0, \sigma_0^2)$ for $\sigma_0^2 = \frac{1}{C}$.

Standard training in the theoretical analysis comprises of full-batch gradient descent on the empirical loss $\mathcal{L}$ with learning rate $\eta \leq \frac{1}{\mathrm{poly}(C)}$ and for $T = \frac{\mathrm{poly}(C)}{\eta}$ iterations.

## C PROOF OF THEOREM 2

We restate Theorem 2, where recall $E$ denotes the teacher ensemble size and $C$ is the number of classes:

**Table 3:** Hyperparameter values used in our theoretical analysis, corresponding to the setup of Allen-Zhu & Li (2020). (*) denotes undefined in our presentation, but appearing in Allen-Zhu & Li (2020), because the hyperparameter takes only one value in this work. We note that $m$ is restricted to be polylog$(C)$ in the setting of Theorem 2.

| Hyperparameter | Description | Value(s) |
|---|---|---|
| $N$ | Training set size | $\frac{C^{1.2}}{\mu}$ |
| $\mu$ | Proportion of single-view data | $\frac{1}{\text{poly}(C)}$ |
| $d$ | Input patch dimension | $\text{poly}(C)$ |
| $P$ | Number of patches per input | $C^2$ |
| $m$ | Number of channels per class | $[\text{polylog}(C), C]$ |
| $s$ | Out-of-class attribute sparsity | $C^{0.2}$ |
| $\sigma_0$ | Parameter initialisation standard deviation (std) | $C^{-0.5}$ |
| $\sigma_p$ | Input patch additive noise std | $\frac{1}{\sqrt{d}\text{polylog}(C)}$ |
| $\Gamma$ | Out-of-class attribute strength in single-view data. | $\frac{1}{\text{polylog}(C)}$ |
| $\varrho$ | $\widetilde{\text{ReLU}}$ threshold | $\frac{1}{\sqrt{d}\text{polylog}(C)}$ |
| q (*) | $\widetilde{\text{ReLU}}$ mid-section exponent | 4 |
| $\rho$ (*) | In-class weaker attribute strength in single-view data | $C^{-0.2}$ |

**Theorem 2** (FKD improves student generalisation and is better with larger ensemble). *Given an arbitrary $\epsilon > 0$. For any ensemble size $E$ of teacher NNs trained as in Theorem 1 and sufficiently many classes $C$, for $m = \text{polylog}(C)$, with learning rate $\eta \leq \frac{1}{\text{poly}(C)}$, and training time $T^* = \frac{\text{poly}(C)}{\eta}$, the ensemble teacher knowledge can be distilled into a single student model $f^{(T^*)}$ using only teacher feature kernel $k_{\mathcal{T}}$, Eq. (7), such that with probability at least $1 - e^{-\Omega(\log^2(C))}$:*

- *Training accuracy is perfect: For all $(\boldsymbol{x}, \boldsymbol{y}) \in \hat{\mathcal{D}}$, $\boldsymbol{y} = \text{argmax}_c f_c^{(T^*)}(\boldsymbol{x})$.*

- *Test accuracy is good: $\mathbb{P}_{(\boldsymbol{x}, \boldsymbol{y}) \sim \mathcal{D}}\left[\boldsymbol{y} \neq \text{argmax}_c f_c^{(T^*)}(\boldsymbol{x})\right] \leq \left(\frac{1}{2^{E+1}} + \epsilon\right)\mu$.*

**Outline of proof of Theorem 2**

1. We first analyse the feature kernel $k$ for a single trained model, in App. C.1.

2. We then extend our results to an ensembled teacher model feature kernel in App. C.2.

3. We next outline our kernel distillation training scheme and provide a key result concerning how the student's parameters become increasingly correlated with the teacher's learnt views in Apps. C.3 and C.4

4. We combine all threads to prove the final result in App. C.5.

## C.1 FEATURE KERNEL FOR A SINGLE TRAINED MODEL

To prove Theorem 2, we first calculate the feature kernel $k_*$ for a single trained model with trained parameters $\boldsymbol{\theta}^*$ from initialisation $\boldsymbol{\theta}^0$. The point of this exercise is to show that the feature kernel $k_*$ is able to detect whether two inputs $\boldsymbol{x}, \boldsymbol{x}'$ share a common attribute which is in the subset of attributes that was learnt by the trained parameters $\boldsymbol{\theta}^*$, due to random correlation with initialised parameters $\boldsymbol{\theta}^0$. This is formally shown in Lemma 1.

Recall the definition of the feature kernel $k$ for our CNN architecture:

$$k(\boldsymbol{x}, \boldsymbol{x}') = \sum_{c=1}^{C} \sum_{r=1}^{m} \sum_{p,p'=1}^{P} \widetilde{\text{ReLU}}(\langle \boldsymbol{\theta}_{c,r}, \boldsymbol{x}_p \rangle) \cdot \widetilde{\text{ReLU}}(\langle \boldsymbol{\theta}_{c,r}, \boldsymbol{x}'_{p'} \rangle)$$

Let us first make a few more useful definitions. For $l = 1, 2$, we have:

**Definition 2.**

$$Z_{c,l}(\boldsymbol{x}) \overset{\text{def}}{=} \mathbb{1}\{v_{c,l} \in \mathcal{V}(\boldsymbol{x})\} \sum_{p \in \mathcal{P}_{v_{c,l}}(\boldsymbol{x})} z_p.$$

> **Intuition:**
>
> $Z_{c,l}(\boldsymbol{x})$ is a scalar constant describing the strength, in $\boldsymbol{x}$, of the presence of attribute $v_{c,l}$ (where $c$ may either correspond to true class $y$ or to a different class; the latter has the effect of producing 'soft labels' when using vanilla knowledge distillation).

**Definition 3.**

$$\Phi^*_{c,l} \overset{\text{def}}{=} \sum_{r \in [m]} [\langle \boldsymbol{\theta}^t_{c,r}, v_{c,l} \rangle]^+.$$

**Definition 4.**

$$\Upsilon^*_{c,l} \overset{\text{def}}{=} \sum_{r \in [m]} [\langle \boldsymbol{\theta}^*_{c,r}, v_{c,l} \rangle]^{+^2}.$$

> **Intuition:**
>
> $\Upsilon^*_{c,l}, \Phi^*_{c,l}$ are both parameter-dependent, data-independent scalars that describe the amount that attribute $v_{c,l}$ has been 'learnt' by the network parameters. $\Upsilon^*_{c,l}$ appears in the feature kernel whereas $\Phi^*_{c,l}$ appears in the function predictions $f_c$ directly, for our particular architecture in Eq. (4) (Allen-Zhu & Li, 2020).

**Lemma 1.** *We have, for $\boldsymbol{x}, \boldsymbol{x}' \sim \hat{\mathcal{D}}_m$,*

$$k_*(\boldsymbol{x}, \boldsymbol{x}') = \sum_{(c,l):(c,3-l) \notin \mathcal{M}} \Upsilon^*_{c,l} Z_{c,l}(\boldsymbol{x}) Z_{c,l}(\boldsymbol{x}') \pm O\left(\frac{s_{\mathcal{M}}(\boldsymbol{x}, \boldsymbol{x}')}{\text{polylog}(C)}\right) + \tilde{O}\left(\frac{1}{C^{0.8}}\right)$$

*where*
$$s_{\mathcal{M}}(\boldsymbol{x}, \boldsymbol{x}') = |\{(c,l) : v_{c,l} \in \mathcal{V}(\boldsymbol{x}) \cap \mathcal{V}(\boldsymbol{x}') \text{ and } (c, 3-l) \notin \mathcal{M}\}|$$

*is the number of shared attributes of $\boldsymbol{x}$ and $\boldsymbol{x}'$, that are also members of the set of attributes learnt by the single trained model with parameters $\boldsymbol{\theta}^*$, and $\mathcal{M}$ is defined in **Fact A.e** below.*

*Proof.* We have

$$k_*(\boldsymbol{x}, \boldsymbol{x}') = \sum_{c=1}^{C} \sum_{r=1}^{m} \Big[ \sum_{p=1}^{P} \widetilde{\text{ReLU}}(\langle \boldsymbol{\theta}^*_{c,r}, \boldsymbol{x}_p \rangle) \cdot \sum_{p'=1}^{P} \widetilde{\text{ReLU}}(\langle \boldsymbol{\theta}^*_{c,r}, \boldsymbol{x}'_{p'} \rangle) \Big].$$

First, define the hidden-layer activations for class $c$ and channel $r$ to be:

**Definition 5.**

$$\Psi_{c,r}(\boldsymbol{x}) \overset{\text{def}}{=} \sum_{p=1}^{P} \widetilde{\text{ReLU}}(\langle \boldsymbol{\theta}^*_{c,r}, \boldsymbol{x}_p \rangle).$$

We know from Theorem C.2. of Allen-Zhu & Li (2020) that, for multi-view $\boldsymbol{x} \in \hat{\mathcal{D}}_m$ and some $c \in [C]$:

**Fact A.a** For every $p \in \mathcal{P}_{v_{c,l}}(\boldsymbol{x})$ and for $l = 1, 2$, we have: $\langle \boldsymbol{\theta}^*_{c,r}, \boldsymbol{x}_p \rangle = \langle \boldsymbol{\theta}^*_{c,r}, v_{c,l} \rangle z_p \pm \tilde{o}(\sigma_0)$

**Fact A.b** For every $p \in \mathcal{P}(\boldsymbol{x}) \backslash (\mathcal{P}_{v_{c,1}}(\boldsymbol{x}) \cup \mathcal{P}_{v_{c,2}}(\boldsymbol{x}))$, we have $|\langle \boldsymbol{\theta}^*_{c,r}, \boldsymbol{x}_p \rangle| \leq \tilde{O}(\sigma_0)$

**Fact A.c** For every $p \in [P] \backslash \mathcal{P}(\boldsymbol{x})$, we have $|\langle \boldsymbol{\theta}^*_{c,r}, \boldsymbol{x}_p \rangle| \leq \tilde{O}(\frac{\sigma_0}{\sqrt{C}})$

**Fact A.d** For every $r \in [m] \backslash \mathcal{M}^0_c$, every $l \in [2]$, it holds that $\langle \boldsymbol{\theta}^*_{c,r}, v_{c,l} \rangle \leq \tilde{O}(\sigma_0)$, where:

$$\mathcal{M}^0_c \stackrel{\text{def}}{=} \left\{ r \in [m] \,\middle|\, \exists l \in [2] : \langle \boldsymbol{\theta}^0_{c,r}, v_{c,l} \rangle \geq (1 - O(\frac{1}{\log(C)})) \cdot \max_{r \in [m]} [\langle \boldsymbol{\theta}^0_{c,r}, v_{c,l} \rangle]^+ \right\}.$$

Note from Proposition B.1 of Allen-Zhu & Li (2020), that $m_0 \stackrel{\text{def}}{=} |\mathcal{M}^0_c| = O(\log^5(C))$ with probability at least $1 - e^{-\Omega(\log^5(C))}$.

> **Intuition:**
> $\mathcal{M}^0_c$ denotes the key channels in $[m]$ which have 'won the lottery' and are relevant for class $c$, in that in the $C \to \infty$ limit the predictions for $f_c$ are the same as if one forgets the other channels, as shown in Allen-Zhu & Li (2020).

**Fact A.e** For every $p \in \mathcal{P}_{v_{c,l}}(\boldsymbol{x})$ and $r \in [m]$, if $(c, 3-l) \in \mathcal{M}$, then we have: $|\langle \boldsymbol{\theta}^*_{c,r}, x_p \rangle| \leq \tilde{O}(\sigma_0)$, where:

$$\mathcal{M} \stackrel{\text{def}}{=} \left\{ (c, l) \in [C] \times [2] \,\middle|\, \max_{r \in [m]} [\langle \boldsymbol{\theta}^0_{c,r}, v_{c,l} \rangle]^+ \geq \left(1 + \frac{1}{\log^2(m)}\right) \cdot \max_{r \in [m]} [\langle \boldsymbol{\theta}^0_{c,r}, v_{c,3-l} \rangle]^+ \right\}$$

> **Intuition:**
> $\mathcal{M}$ denotes the data attributes $v_{c,l}$ which are more likely to be learnt by the NN parameters (compared to their fellow class attributes $v_{c,3-l}$) because of correlations of the initial parameters with such attributes. So, **Fact A.e** is saying that if $(c, 3 - l) \in \mathcal{M}$, then the attribute $(c, l)$ is not learnt at all during standard single model training.

Thus,

$$\Psi_{c,r}(\boldsymbol{x}) = \sum_{l=1}^{2} \mathbb{1}\{v_{c,l} \in \mathcal{V}(\boldsymbol{x})\} \cdot \sum_{p \in \mathcal{P}_{v_{c,l}}(\boldsymbol{x})} \widetilde{\text{ReLU}}\big(\langle \boldsymbol{\theta}^*_{c,r}, \boldsymbol{x}_p \rangle\big) \tag{13}$$

$$+ \sum_{p \in \mathcal{P}(\boldsymbol{x}) \backslash \bigcup_l \{\mathcal{P}_{v_{c,l}}(\boldsymbol{x})\}} \widetilde{\text{ReLU}}\big(\tilde{O}(\sigma_0)\big) \tag{14}$$

$$+ \sum_{p \in [P] \backslash \mathcal{P}(\boldsymbol{x})} \widetilde{\text{ReLU}}\big(\tilde{O}(\frac{\sigma_0}{\sqrt{C}})\big). \tag{15}$$

First, note that $|\mathcal{P}_{v_{c',l'}}(\boldsymbol{x})| = C_p$ is constant $\forall c', l'$. From **Fact A.b**, Eq. (14) can be easily seen to be $\tilde{O}(\sigma_0^4 s) = \tilde{O}(C^{-1.8})$ as $\sigma_0^2 = \frac{1}{C}$ and $s = C^{0.2}$, and likewise Eq. (15) can be seen to be $\tilde{O}((\frac{\sigma_0}{\sqrt{C}})^4 P) = \tilde{O}(C^{-2})$ as $P = C^2$, by **Fact A.c**. We note here that summing these equations over $m$ and $C$ will be bounded above by $O(\frac{1}{\text{polylog}(C)})$.

Now, let's consider Eq. (13). Let notation $v_{c,1}, v_{c,2} \in S$ denote either $v_{c,1}$ or $v_{c,2} \in S$ for some set $S$, and $v_{c,1}, v_{c,2} \notin S$ denote neither $v_{c,1}$ nor $v_{c,2} \in S$.

There are three cases to consider:

1. If $v_{c,1}, v_{c,2} \notin \mathcal{V}(\boldsymbol{x})$, then Eq. (13) is zero.

2. Else if $\exists l \in [2]$ such that $v_{c,l} \in \mathcal{V}(\boldsymbol{x})$ and $(c, 3-l) \in \mathcal{M}$, then by **Fact A.e**, we have Eq. (13) is $\tilde{O}(\sigma_0^4) = \tilde{O}(C^{-2})$.

> **Intuition:**
>
> This setting is when only attribute $(c, l)$ appears in $\boldsymbol{x}$, but attribute $(c, 3-l)$ was dominant at initialisation so $(c, l)$ has not been learnt by $\boldsymbol{\theta}^*$.

3. Else, we have that:

$$\text{Eq. (13)} = \sum_{l=1}^{2} \mathbb{1}\{v_{c,l} \in \mathcal{V}(\boldsymbol{x})\} \cdot \sum_{p \in \mathcal{P}_{v_{c,l}}(\boldsymbol{x})} \widetilde{\text{ReLU}}\big(\langle \boldsymbol{\theta}_{c,r}^*, \boldsymbol{x}_p \rangle\big).$$

Putting this all together with **Fact A.a**, we see that

$$\Psi_{c,r}(\boldsymbol{x}) = \sum_{l=1}^{2} \mathbb{1}\{(c, l) \in s_{\mathcal{M}}(\boldsymbol{x}, \boldsymbol{x})\} \sum_{p \in \mathcal{P}_{v_{c,l}}(\boldsymbol{x})} \widetilde{\text{ReLU}}\big(\langle \boldsymbol{\theta}_{c,r}^*, v_{c,l} \rangle z_p + \tilde{o}(\sigma_0)\big) + \tilde{O}\big(\frac{1}{C^{1.8}}\big).$$

Now, if $r \notin \mathcal{M}_c^0$ (i.e. neuron $r$ is not dominant at initialisation), and $(c, 1), (c, 2) \in s_{\mathcal{M}}(\boldsymbol{x}, \boldsymbol{x})$, the by **Fact A.d**, we have that $\Psi_{c,r}(\boldsymbol{x}) = \tilde{O}(C^{-1.8})$.

On the other hand, recall $\varrho = \frac{1}{\text{polylog}(C)}$ and $m_0 = O(\log^5(C))$. And also, recall that if $z > \varrho$, then $\widetilde{\text{ReLU}}(z) = z + O(\varrho)$. Moreover, by Claim C.11 of Allen-Zhu & Li (2020) we know that $\exists r', l'$ such that $\langle \boldsymbol{\theta}_{c,r'}^*, v_{c,l'} \rangle = \Omega(\frac{1}{m_0})$. Hence, for any $r \in \mathcal{M}_c^0$, we know that either:

1. $\Psi_{c,r}(\boldsymbol{x}) = \sum_{l=1}^{2} \mathbb{1}\{(c, l) \in s_{\mathcal{M}}(\boldsymbol{x}, \boldsymbol{x})\}\big(\langle \boldsymbol{\theta}_{c,r}^*, v_{c,l} \rangle Z_{c,l}(\boldsymbol{x}) \pm O(\varrho)\big) + \tilde{O}\big(\frac{1}{C^{1.8}}\big)$, or

2. $\Psi_{c,r}(\boldsymbol{x}) = \mathbb{1}\{(c, 1), (c, 2) \in s_{\mathcal{M}}(\boldsymbol{x}, \boldsymbol{x})\} O(\varrho) + \tilde{O}\big(\frac{1}{C^{1.8}}\big)$.

Moreover, $\Psi_{c,r}(\boldsymbol{x}) = \tilde{O}(1) \ \forall r$, by e.g. Lemma C.21 of Allen-Zhu & Li (2020). And so it can be seen, for $\varrho$ small enough we have:

$$\sum_{r=1}^{m} \Psi_{c,r}(\boldsymbol{x}) \Psi_{c,r}(\boldsymbol{x}')$$

$$= \sum_{l=1}^{2} \mathbb{1}\{(c, 3-l) \notin \mathcal{M}\} Z_{c,l}(\boldsymbol{x}) Z_{c,l}(\boldsymbol{x}')\big[\sum_{r=1}^{m} \big[\langle \boldsymbol{\theta}_{c,r}^*, v_{c,l} \rangle\big]^{+^2} \pm O\big(\frac{1}{\text{polylog}(C)}\big)\big] + \tilde{O}\big(\frac{1}{C^{1.8}}\big)$$

$$= \sum_{l=1}^{2} \mathbb{1}\{(c, 3-l) \notin \mathcal{M}\} Z_{c,l}(\boldsymbol{x}) Z_{c,l}(\boldsymbol{x}')\big[\Upsilon_{c,l}^* \pm O\big(\frac{1}{\text{polylog}(C)}\big)\big] + \tilde{O}\big(\frac{1}{C^{1.8}}\big)$$

where we also use **Fact A.e** above, such that it is not possible for both $\langle \boldsymbol{\theta}_{c,r}^*, v_{c,l} \rangle^+$ and $\langle \boldsymbol{\theta}_{c,r}^*, v_{c,3-l} \rangle^+$ to be large.[8] Note also from e.g. the proof of Allen-Zhu & Li (2020) Theorem 1, that if $(c, 3 - l) \notin \mathcal{M}$, then $\max_r \langle \boldsymbol{\theta}_{c,r}^*, v_{c,l} \rangle = \tilde{\Theta}(1)$.

Thus, we see that the contribution to $k$ from the $m$ class-$c$ channels/neurons is:

$$k_c(\boldsymbol{x}, \boldsymbol{x}') \stackrel{\text{def}}{=} \sum_{r=1}^m \Psi_{c,r}(\boldsymbol{x}) \Psi_{c,r}(\boldsymbol{x}') \tag{16}$$

$$= \begin{cases} \tilde{\Theta}(1) & \text{if } (c,1), (c,2) \in s_{\mathcal{M}}(\boldsymbol{x}, \boldsymbol{x}') \\ \tilde{O}(\frac{1}{C^{1.8}}) & \text{else} \end{cases} \tag{17}$$

Summing $k_c$ over $[C]$ completes the proof of the lemma. $\qquad\square$

We now use Lemma 1 to analyse $k_*$ and $\rho_*(\boldsymbol{x}, \boldsymbol{x}') = \frac{k_*(\boldsymbol{x}, \boldsymbol{x}')}{\sqrt{k_*(\boldsymbol{x}, \boldsymbol{x}) k_*(\boldsymbol{x}', \boldsymbol{x}')}}$.

First we look at $\Upsilon_{c,l}^*$. Recall that $\langle \boldsymbol{\theta}_{c,r}^*, v_{c,l} \rangle = \tilde{O}(1)$ for all $c \in [C], l \in [2], r \in [m]$, and likewise so is $\mathcal{M}_c^0$.

More specifically,

- If $(c, l) \in \mathcal{M}$, then we have from the proof of Theorem 1 in Allen-Zhu & Li (2020) that $\sum_r \langle \boldsymbol{\theta}_{c,r}^*, v_{c,l} \rangle^+ \geq \Omega(\log(C))$, and so $\max_r \langle \boldsymbol{\theta}_{c,r}^*, v_{c,l} \rangle^+ = \tilde{\Theta}(1)$.
- If neither $(c, 1)$ nor $(c, 2)$ are in $\mathcal{M}$, then Claim C.10 of Allen-Zhu & Li (2020) shows us that both $\max_r \langle \boldsymbol{\theta}_{c,r}^*, v_{c,1} \rangle^+, \max_r \langle \boldsymbol{\theta}_{c,r}^*, v_{c,2} \rangle^+$ are $\tilde{\Theta}(1)$.

Combining these facts, we have that:

$$\Upsilon_{c,l}^* = \tilde{\Theta}(1) \qquad \text{if } (c, 3 - l) \notin \mathcal{M}$$

and so from Lemma 1, we have:

$$k_*(\boldsymbol{x}, \boldsymbol{x}') = \sum_{(c,l):(c,3-l) \notin \mathcal{M}} \Upsilon_{c,l}^* Z_{c,l}(\boldsymbol{x}) Z_{c,l}(\boldsymbol{x}') \pm O\left(\frac{s_{\mathcal{M}}(\boldsymbol{x}, \boldsymbol{x}')}{\text{polylog}(C)}\right) + \tilde{O}(C^{-0.8}) \tag{18}$$

$$= \tilde{\Theta}(1) \sum_{(c,l):(c,3-l) \notin \mathcal{M}} Z_{c,l}(\boldsymbol{x}) Z_{c,l}(\boldsymbol{x}') \pm O\left(\frac{s_{\mathcal{M}}(\boldsymbol{x}, \boldsymbol{x}')}{\text{polylog}(C)}\right) + \tilde{O}(C^{-0.8}) \tag{19}$$

$$= \tilde{\Theta}(s_{\mathcal{M}}(\boldsymbol{x}, \boldsymbol{x}')) \pm O\left(\frac{s_{\mathcal{M}}(\boldsymbol{x}, \boldsymbol{x}')}{\text{polylog}(C)}\right) + \tilde{O}(C^{-0.8}). \tag{20}$$

Finally, we arrive at an expression for the correlation kernel $\rho_*$ of a trained single model by

$$\rho_*(\boldsymbol{x}, \boldsymbol{x}') = \tilde{\Theta}\left(\frac{s_{\mathcal{M}}(\boldsymbol{x}, \boldsymbol{x}')}{\sqrt{s_{\mathcal{M}}(\boldsymbol{x}) s_{\mathcal{M}}(\boldsymbol{x}')}}\right)\left(1 + O\left(\frac{1}{\text{polylog}(C)}\right)\right) + \tilde{O}\left(\frac{1}{C^{0.8}}\right) \tag{21}$$

where we define that $s_{\mathcal{M}}(\boldsymbol{x}) = s_{\mathcal{M}}(\boldsymbol{x}, \boldsymbol{x})$ is the number of attributes in $\boldsymbol{x}$ that have been learnt by the trained network $\boldsymbol{\theta}^*$, and is $s(1 \pm o(1))$ with high probability.

---

[8]Unless neither $(c, 1)$ nor $(c, 2) \in \mathcal{M}$ for a given $c$, but that only occurs in $o(C)$ classes, and does not change the order of e.g. $s_{\mathcal{M}}(\boldsymbol{x}, \boldsymbol{x}')$ which is what we really care about.

> **Intuition:**
>
> Compare $\rho_*$ in Eq. (21) to the 'soft' probability labels $p^\tau \in \mathbb{R}^C$ of Allen-Zhu & Li (2020) (Claim F.4) with temperature $\tau = \frac{1}{\log^2(C)}$:
>
> $$p_c^\tau(\boldsymbol{x}) = \begin{cases} \frac{1}{s(\boldsymbol{x})} & \text{if } v_{c,1} \text{ or } v_{c,2} \text{ is in } \mathcal{V}(\boldsymbol{x}) \\ 0 & \text{else} \end{cases}$$
>
> where $s(\boldsymbol{x})$ is the number of indices $c \in [C]$ such that $v_{c,1}$ or $v_{c,2}$ is in $\mathcal{V}(\boldsymbol{x})$. Note, the setting of Allen-Zhu & Li (2020) is with a large $\tilde{\Omega}(1)$ ensemble, so every attribute is learnt (akin to $\mathcal{M}$ being empty for us). In the case that $\mathcal{M} = \{\}$ being empty, if $\boldsymbol{x} \neq \boldsymbol{x}'$ and they share at least one feature, then from App. B.1.1 with high probability they will share exactly one feature, so that $s_{\mathcal{M}}(\boldsymbol{x}, \boldsymbol{x}') = 1$. Moreover, $s(\boldsymbol{x}) = s_{\mathcal{M}}(\boldsymbol{x}) \approx s_{\mathcal{M}}(\boldsymbol{x}')$, hence we see that Eq. (21) matches roughly with $p_c^\tau(\boldsymbol{x})$, but without the need for a temperature hyperparameter.
>
> We see that vanilla KD learns new attributes by comparing a single data point $\boldsymbol{x}$ between classes, and giving larger target labels to the classes where ambiguous attributes learnt by the teacher are present in $\boldsymbol{x}$. On the other hand, $\rho_*$ gives higher values to data pairs $\boldsymbol{x}, \boldsymbol{x}'$ that share attributes that have been learnt by the trained model, and as we will see later this is how FKD learns new attributes in the student.

### C.2 ENSEMBLED TEACHER

To summarise what we have done so far in App. C.1, we have seen in Eqs. (20) and (21) that it is possible, for a single trained model $\boldsymbol{\theta}^*$, to simplify both the feature kernel $k_*(\boldsymbol{x}, \boldsymbol{x}')$ and correlation kernel $\rho_*(\boldsymbol{x}, \boldsymbol{x}')$ in terms of the number of shared attributes between $\boldsymbol{x}, \boldsymbol{x}'$ which are also learnt by the trained model. The set of attributes learnt by the single trained model is captured by the set

$$\mathcal{M} = \left\{ (c,l) \in [C] \times [2] \,\middle|\, \max_{r \in [m]} [\langle \boldsymbol{\theta}_{c,r}^0, v_{c,l} \rangle]^+ \geq \left(1 + \frac{1}{\log^2(m)}\right) \cdot \max_{r \in [m]} [\langle \boldsymbol{\theta}_{c,r}^0, v_{c,3-l} \rangle]^+ \right\}$$

where $\boldsymbol{\theta}_0$ was the random parameter initialisation for $\boldsymbol{\theta}^*$. From **Fact A.e**, we know that if $(c, 3-l) \in \mathcal{M}$, then the attribute $v_{c,l}$ has not been learnt by the network.

Consider now an ensemble of $E = \Theta(1)$ independently trained networks, $\{\boldsymbol{\theta}_e^*\}_{e=1}^E$, with an averaged feature kernel:

$$k_{\mathcal{T}}(\boldsymbol{x}, \boldsymbol{x}') = \frac{1}{E} \sum_{e=1}^E k_e^*(\boldsymbol{x}, \boldsymbol{x}').$$

Suppose $\{\boldsymbol{\theta}^{e,0}\}_{e=1}^E$ denotes the corresponding independent parameter initialisations. Then, for each $e \in [E]$, let us define:

$$\mathcal{M}_e \overset{\text{def}}{=} \left\{ (c,l) \in [C] \times [2] \,\middle|\, \max_{r \in [m]} [\langle \boldsymbol{\theta}_{c,r}^{e,0}, v_{c,l} \rangle]^+ \geq \left(1 + \frac{1}{\log^2(m)}\right) \cdot \max_{r \in [m]} [\langle \boldsymbol{\theta}_{c,r}^{e,0}, v_{c,3-l} \rangle]^+ \right\}.$$

Note that these $\mathcal{M}_e$ are completely independent sets due to the independent initialisations, and also by Proposition B.2 of Allen-Zhu & Li (2020), we know that

$$\mathbb{P}\big[(c,1) \text{ or } (c,2) \in \mathcal{M}_e\big] \geq 1 - o(1) \qquad \forall c \in [C], e \in [E].$$

Therefore,

$$C \geq |\mathcal{M}_e| \geq C(1 - o_p(1)) \qquad \forall e \in [E].$$

Moreover, Eq. (11) & Proposition B.2 of Allen-Zhu & Li (2020) tell us that each of the two attributes $v_{c,1}, v_{c,2}$ are equally likely to be in $\mathcal{M}_e$ (and so learnt in the multi-view setup), This means that:

$$|\bigcap_{e=1}^{E} \mathcal{M}_e| = \frac{1}{2^{E-1}} C(1 - o_p(1)).$$

Define $\mathcal{M}_{\mathcal{T}} = \bigcap_{e=1}^{E} \mathcal{M}_e$. From Eq. (19) and the definition of $k_{\mathcal{T}}$, we see that:

$$k_{\mathcal{T}}(\boldsymbol{x}, \boldsymbol{x}') = \tilde{\Theta}(1) \sum_{(c,l)} \mathbb{1}\{(c, 3-l) \notin \mathcal{M}_{\mathcal{T}}\} Z_{c,l}(\boldsymbol{x}) Z_{c,l}(\boldsymbol{x}') \pm O\left(\frac{s_{\mathcal{M}_{\mathcal{T}}}(\boldsymbol{x}, \boldsymbol{x}')}{\text{polylog}(C)}\right) + \tilde{O}(C^{-0.8}) \tag{22}$$

$$= \tilde{\Theta}(s_{\mathcal{M}_{\mathcal{T}}}(\boldsymbol{x}, \boldsymbol{x}')) \pm O\left(\frac{s_{\mathcal{M}_{\mathcal{T}}}(\boldsymbol{x}, \boldsymbol{x}')}{\text{polylog}(C)}\right) + \tilde{O}(C^{-0.8}) \tag{23}$$

where for the reader's convenience, we redefine:

$$s_{\mathcal{M}_{\mathcal{T}}}(\boldsymbol{x}, \boldsymbol{x}') = \{(c,l) : \ v_{c,l} \in \mathcal{V}(\boldsymbol{x}) \cap \mathcal{V}(\boldsymbol{x}') \text{ and } (c, 3-l) \notin \mathcal{M}_{\mathcal{T}}\}.$$

> **Intuition:**
>
> We see that only for those attributes $(c, l)$ such that $(c, 3-l) \in \mathcal{M}_{\mathcal{T}}$ does the ensembled teacher $k_{\mathcal{T}}$ miss the fact that we should have a strong $\tilde{\Theta}(1)$ kernel value between $\boldsymbol{x}, \boldsymbol{x}'$. This is when $|\mathcal{V}(\boldsymbol{x}) \cap \mathcal{V}(\boldsymbol{x}')| = \{v_{c,l}\}$ is non-empty (or in other words, when $s_{\mathcal{M}_{\mathcal{T}}}(\boldsymbol{x}, \boldsymbol{x}') \neq |\mathcal{V}(\boldsymbol{x}) \cap \mathcal{V}(\boldsymbol{x}')|$), and hence there should be a large kernel value $k_{\mathcal{T}}(\boldsymbol{x}, \boldsymbol{x}')$.
>
> So we see that only $|\mathcal{M}_{\mathcal{T}}|$ of the attributes are not learnt by the teacher, which is a fraction $\frac{|\mathcal{M}_{\mathcal{T}}|}{2C} = \frac{1}{2^E}(1 - o(1))$ of all the attributes. These missed attributes are where the $\frac{1}{2^{E+1}}$ test error in Theorem 2 comes from (teacher ensemble of size E, and plus 1 for the attributes learnt from the student's initialisation too).

What's more, we can decompose the teacher's feature kernel $k_{\mathcal{T}} = \sum_c k_c^{\mathcal{T}}$ into contributions $k_c^{\mathcal{T}}$ from each class $c$, like in Eq. (16). From Eq. (17), we see that the contribution to the teacher's feature kernel from class , for $\boldsymbol{x} \neq \boldsymbol{x}'$;

$$k_c^{\mathcal{T}}(\boldsymbol{x}, \boldsymbol{x}') = \begin{cases} \tilde{\Theta}(1) & \text{if } (c,1), (c,2) \in s_{\mathcal{M}_{\mathcal{T}}}(\boldsymbol{x}, \boldsymbol{x}') \\ \tilde{O}(\frac{1}{C^{1.8}}) & \text{else,} \end{cases} \tag{24}$$

is able to decipher between whether or not $\boldsymbol{x}, \boldsymbol{x}'$ share an attribute from class $c$ for all attributes apart from those $(c, l)$ such that $(c, 3-l) \in \mathcal{M}_{\mathcal{T}}$.

## C.3 TRAINING SCHEME FOR FKD

We note at this point that we are morally done in terms of proving Theorem 2, with Eqs. (23) and (24), our key results telling us that the (ensemble) teacher kernel $k_{\mathcal{T}}$ can identify when two inputs share common attributes that have been learnt by the (ensemble) teacher, and more specifically that $k_c^{\mathcal{T}}$ can do so when said common attribute is from class $c$.

What remains is a repackaging of the proof techniques of Allen-Zhu & Li (2020) (particularly for their Theorem 4 regarding self-distillation), that knowledge distillation (this time only using feature kernels instead of temperature-scaled logits, and with explicit dependence on teacher ensemble size) can improve generalisation performance of a student.

For convenience, the theoretical analysis of Allen-Zhu & Li (2020) introduces some slight discrepancies between the actual practical weight updates of vanilla KD Hinton et al. (2015), i.e. the gradients of:

$$\tilde{\mathcal{L}} = \mathcal{L} + \lambda \frac{1}{N} \sum_i L(\frac{f(\boldsymbol{x}_i)}{\tau}, \frac{f_{\mathcal{T}}(\boldsymbol{x}_i)}{\tau}),$$

and the weight updates in their theoretical exposition. Namely,

1. The authors assume that a temperature-dependent threshold caps the logits to give soft labels:
$$p_c^\tau(\boldsymbol{x}) = \frac{e^{\min\{\tau^2 f_c(\boldsymbol{x}), 1\}/\tau}}{\sum_{j \in [C]} e^{\min\{\tau^2 f_j(\boldsymbol{x}), 1\}/\tau}}.$$

2. The authors truncate the negative part of the gradient of the KD regularisation to only encourage logits to increase not decrease, with weight updates for $\boldsymbol{\theta}_{c,r}$ on input $\boldsymbol{x}$:

$$-\Delta\boldsymbol{\theta}_{c,r}^t \overset{\text{def}}{=} \boldsymbol{\theta}_{c,r}^t - \boldsymbol{\theta}_{c,r}^{t+1} \propto \nabla_{\boldsymbol{\theta}_{c,r}} \mathcal{L} + \eta \frac{1}{N} \sum_i \left(p_c^\tau(\boldsymbol{x}) - p_c^{\tau,\mathcal{T}}(\boldsymbol{x})\right)^- \nabla_{\boldsymbol{\theta}_{c,r}} f_c(\boldsymbol{x})$$

where $p^{\tau,\mathcal{T}}$ are the temperature-scaled teacher labels.

3. The authors scale the output of both student and teacher models by a (polylogarithmic) factor, in order to ensure that both reach the threshold to give soft labels in Item 1. above.

4. Self-distillation (Furlanello et al., 2018; Zhang et al., 2019) distils a single teacher and a single student of same architecture into the student, like an ensemble of size 2 (student+teacher). Allen-Zhu & Li (2020) modify the training scheme for their theoretical analysis of self-distillation so that the student is first trained on its own in order separate learning its own attributes/features from those of the teacher. Our analysis covers a similar scheme.

These modifications are justified in that they make the theoretical analysis more convenient, whilst illustrating the main mechanisms by which KD works, which is to share 'dark knowledge' that is held in the teacher (in the form of the multi-view attributes that have been acquired by the teacher due to its parameter initialisation), with the student.

In the same vein, we now introduce some modifications to the practical implementation of FKD we propose in Alg. 1 to aid our theoretical analysis, and describe the main mechanisms by which FKD works, corroborating our initial analyses in Section 2 and App. A about how the feature kernel is a crucial object in any NN and captures all the 'dark knowledge' that a teacher network could possess in the multi-view data setting.

It is likely possible to extend our proof of Theorem 2 with different modifications/training schemes, but given that the focus of this work is to introduce FKD as a principled alternative to vanilla KD with certain advantages such as prediction-space independence, and that the multi-view setting we consider is a plausible simplification of real world data (as demonstrated in Allen-Zhu & Li (2020)), we leave this to future work. We stress that any simplifications to the update rule in Alg. 1 for this section can be efficiently computed, only requiring access to pairwise evaluations of the student and teacher feature kernels, if need be.

**Modified training regime for FKD**

1. We first suppose that the student is trained as standard (as in App. B.4) for $T_1 = \frac{\text{poly}(C)}{\eta}$ steps, and learns its own subset of attributes $\mathcal{M}_{\mathcal{S}}$, dependent on its initialisation $\boldsymbol{\theta}_s^0$, before being trained with the FKD objective:

> **Intuition:**
>
> This mirrors the self-distillation setup of Allen-Zhu & Li (2020) Theorem 4. The idea being that the student first learns $\mathcal{M}_\mathcal{S}$ before picking up the other attributes that the teacher has access to.

2. For a given feature kernel $k$, we threshold the feature kernel $k$ based on value, to define a modification $\tilde{k}$ such that

$$\tilde{k}(\boldsymbol{x}, \boldsymbol{x}') = \begin{cases} 1 & \text{if } k(\boldsymbol{x}, \boldsymbol{x}') \geq \frac{1}{m^2} \\ 0 & \text{else} \end{cases}$$

> **Intuition:**
>
> This condition delineates between the setting where $\boldsymbol{x}, \boldsymbol{x}'$ share common attributes learnt by student parameters $\boldsymbol{\theta}^{T_1}$ in the initial phase of training (i.e. delineates between whether $s_{\mathcal{M}_s}(\boldsymbol{x}, \boldsymbol{x}')$ nonempty or empty).
>
> To see this: note that if $v_{c,l} \in \mathcal{V}(\boldsymbol{x}) \cap \mathcal{V}(\boldsymbol{x}')$ and $(c, 3-l) \notin \mathcal{M}_\mathcal{S}$ then we know from Allen-Zhu & Li (2020) that $\Phi_{c,l}^{T_1} \geq \Omega(\log(C))$.
> Hence $\max_r \langle \boldsymbol{\theta}_{c,r}^*, v_{c,l} \rangle \geq \Omega(\log^{-4}(C))$ as the number of active neurons $m_0 = |\mathcal{M}_c^0| = O(\log^5 C)$, and so it's easy to see that for large enough $m$ we have $k(\boldsymbol{x}, \boldsymbol{x}') \geq \frac{1}{m^2}$ via Lemma 1.
> On the other hand, if $\{v_{c,l}\} = \mathcal{V}(\boldsymbol{x}) \cap \mathcal{V}(\boldsymbol{x}')$ and $(c, 3-l) \in \mathcal{M}_\mathcal{S}$ then from Lemma 1 we know that $k(\boldsymbol{x}, \boldsymbol{x}') = \tilde{O}(C^{-0.8}) \ll \frac{1}{m^2}$.

3. Similar to Allen-Zhu & Li (2020), we also truncate our FKD regularisation to only encourage kernel values to increase, and not decrease. For any input pair $\boldsymbol{x}_1, \boldsymbol{x}_2$, we have parameter update:

$$-\Delta\boldsymbol{\theta}_{c,r}(\boldsymbol{x}_1, \boldsymbol{x}_2) \propto \left( \tilde{k}_c(\boldsymbol{x}_1, \boldsymbol{x}_2) - \tilde{k}_c^\mathcal{T}(\boldsymbol{x}_1, \boldsymbol{x}_2) \right)^- \Big[ \sum_{j \in \{1,2\}} \Psi_{c,r}(\boldsymbol{x}_j) \nabla_{\boldsymbol{\theta}_{c,r}} \Psi_{c,r}(\boldsymbol{x}_{3-j}) \Big]$$

(25)

where recall

$$\Psi_{c,r}(\boldsymbol{x}) \overset{\text{def}}{=} \sum_{p=1}^P \widetilde{\text{ReLU}}(\langle \boldsymbol{\theta}_{c,r}, \boldsymbol{x}_p \rangle) \quad \text{so that} \quad \nabla_{\boldsymbol{\theta}_{c,r}} \Psi_{c,r}(\boldsymbol{x}) = \sum_{p=1}^P \widetilde{\text{ReLU}}'(\langle \boldsymbol{\theta}_{c,r}, \boldsymbol{x}_p \rangle) \boldsymbol{x}_p$$

and

$$k_c(\boldsymbol{x}, \boldsymbol{x}') \overset{\text{def}}{=} \sum_{r=1}^m \Psi_{c,r}(\boldsymbol{x}) \Psi_{c,r}(\boldsymbol{x}')$$

were defined in Definition 5 and Eq. (16).

> **Intuition:**
>
> If the loss was
> $$(k_c(\boldsymbol{x}_1, \boldsymbol{x}_2) - k_c^\mathcal{T}(\boldsymbol{x}_1, \boldsymbol{x}_2))^2$$
> then the gradient with respect to $\boldsymbol{\theta}_{c,r}$ would be:
> $$\left( k_c(\boldsymbol{x}_1, \boldsymbol{x}_2) - k_c^\mathcal{T}(\boldsymbol{x}_1, \boldsymbol{x}_2) \right) \Big[ \sum_{j \in \{1,2\}} \Psi_{c,r}(\boldsymbol{x}_j) \nabla_{\boldsymbol{\theta}_{c,r}} \Psi_{c,r}(\boldsymbol{x}_{3-j}) \Big]$$
> so the only differences with Eq. (25) are truncating $\left( \tilde{k}_c(\boldsymbol{x}_1, \boldsymbol{x}_2) - \tilde{k}_c^\mathcal{T}(\boldsymbol{x}_1, \boldsymbol{x}_2) \right)^-$ and also the thresholding to obtain $\tilde{k}$.

To summarise, after training the student on its own for $T_1$ steps (such that we are in the setting of Theorem 1) to reach parameters $\boldsymbol{\theta}_s^{T_1}$, we update for $T_2 = \frac{\text{poly}(C)}{\eta}$ steps as (hiding $\mathcal{S}$ subscript):

$$\Delta\boldsymbol{\theta}_{c,r}^t = -\eta\mathbb{E}_{\boldsymbol{x}_1,\boldsymbol{x}_2\sim\hat{\mathcal{D}}^2}\big[\big(\tilde{k}_c(\boldsymbol{x}_1,\boldsymbol{x}_2)-\tilde{k}_c^{\mathcal{T}}(\boldsymbol{x}_1,\boldsymbol{x}_2)\big)^-\sum_{j\in\{1,2\}}\Psi_{c,r}(\boldsymbol{x}_j)\nabla_{\boldsymbol{\theta}_{c,r}}\Psi_{c,r}(\boldsymbol{x}_{3-j})\big] \quad (26)$$

### C.4 Feature correlation growths

We now seek to analyse to what extent the attributes $\{v_{c,l}\}_{c,l}$ are learnt during our $T_2$ FKD training steps. The central objects describing how much $v_{c,l}$ has been learnt by parameters $\boldsymbol{\theta}$ are:

$$\Phi_{c,l}^t \overset{\text{def}}{=} \sum_{r\in[m]}[\langle\boldsymbol{\theta}_{c,r}^t,v_{c,l}\rangle]^+ \qquad \text{and} \qquad \Phi_c^t \overset{\text{def}}{=} \sum_{l\in[2]}\Phi_{c,l}^t$$

as well as $\Psi_{c,r}(\boldsymbol{x})$ as defined above.

> **Intuition:**
>
> $\Phi_{c,l}$ is a data-independent quantity that reflects the strength of correlation with feature $v_{c,l}$ by parameters $\boldsymbol{\theta}$. On the other hand, $\Psi_{c,r}(\boldsymbol{x})$ is a data-dependent quantity that reflects the activation of channel $r$ for class $c$ with input $\boldsymbol{x}$.

Also recall that

$$Z_{c,l}(\boldsymbol{x}) \overset{\text{def}}{=} \mathbb{1}\{v_{c,l}\in\mathcal{V}(\boldsymbol{x})\}\sum_{p\in\mathcal{P}_{v_{c,l}}(\boldsymbol{x})}z_p$$

and define $V_{c,r,l}(\boldsymbol{x})$ (which is convenient for calculating the size of gradient updates for $\boldsymbol{\theta}_{c,r}$):

**Definition 6.**

$$V_{c,r,l}(\boldsymbol{x}) \overset{\text{def}}{=} \mathbb{1}\{v_{c,l}\in\mathcal{V}(\boldsymbol{x})\}\sum_{p\in\mathcal{P}_{v_{c,l}}(\boldsymbol{x})}\widetilde{\text{ReLU}}'(\langle\boldsymbol{\theta}_{c,r},\boldsymbol{x}_p\rangle)z_p$$

Like how Lemma 1 simplified the feature kernel in terms of data-dependent $Z_{c,l}(\boldsymbol{x})$ and data-independent $\Upsilon_{c,l}$, we have a result from Allen-Zhu & Li (2020) to simplify function predictions $f_c$ in terms of $Z_{c,l}(\boldsymbol{x})$ and $\Phi_{c,l}$:

**Claim 1** (Claim F.7 from Allen-Zhu & Li (2020))**.** *For every $t \leq T_1 + T_2$, every $c \in [C]$, every $(\boldsymbol{x},\boldsymbol{y}) \in \hat{\mathcal{D}}$ (or every test sample $(\boldsymbol{x},\boldsymbol{y}) \sim \mathcal{D}$ with probability $1 - e^{-\Omega(\log^2(C))}$):*

$$f_c^t(\boldsymbol{x}) = \sum_{l\in[2]}\big(\Phi_{c,l}^t\times Z_{c,l}^t(\boldsymbol{x})\big)\pm O\big(\frac{1}{\text{polylog}(C)}\big)$$

We also have the following facts from Allen-Zhu & Li (2020) regarding the correlation of gradient $\nabla_{\boldsymbol{\theta}_{c,r}}\Psi_{c,r}^t(\boldsymbol{x})$ with $v_{c,l}$ for $(\boldsymbol{x},\boldsymbol{y}) \in \hat{\mathcal{D}}, l \in [2]$ and $r \in [m]$:

**Claim 2** (c.f. Claim F.6 of Allen-Zhu & Li (2020))**.** *For every $t \leq T_1 + T_2$, for every $(\boldsymbol{x},\boldsymbol{y}) \in \hat{\mathcal{D}}$, every $c \in [C]$, $r \in [m]$ and $l \in [2]$:*

- *If $v_{c,1}, v_{c,2} \in \mathcal{V}(\boldsymbol{x})$, then $\langle\nabla_{\boldsymbol{\theta}_{c,r}}\Psi_{c,r}^t(\boldsymbol{x}),v_{c,l}\rangle \geq \big(V_{c,r,l}(\boldsymbol{x})-\tilde{O}(\sigma_p P)\big)$*

- $\langle\nabla_{\boldsymbol{\theta}_{c,r}}\Psi_{c,r}^t,v_{c,l}\rangle \leq \big(\mathbb{1}\{(v_{c,l}\in\mathcal{V}(\boldsymbol{x})\}V_{c,r,l}(\boldsymbol{x})+\tilde{O}(C^{-2})\big)$

- *For every $i \neq c$, $|\langle-\nabla_{\boldsymbol{\theta}_{c,r}}\Psi_{c,r}^t(\boldsymbol{x}),v_{i,l}\rangle| \leq \tilde{O}(C^{-1.5})$*

**Disclaimer** Technically, Allen-Zhu & Li (2020) only show Claims 1 and 2 for $t \leq T_1$ and one would need to use similar proof techniques (such as their inductive hypothesis F.1) to show the case for $T_1 \leq t \leq T_2$, which we skip for conciseness.

We now study the growth of the student's $\Phi_{c,l}$, for those $(c, l)$ which have been learnt by the teacher but not the student:

**Lemma 2** (Correlation Growth for attributes learnt by teacher). *For every* $c \in [C], l \in [2]$, $T_2 \geq t \geq T_1$, *such that* $(c, 3 - l) \notin \mathcal{M}_{\mathcal{T}}$, *suppose* $\Phi_{c,l}^t \leq \frac{1}{2m}$, *then we have:*

$$\Phi_{c,l}^{t+1} \geq \Phi_{c,l}^t + \tilde{\Omega}(\frac{\eta s^2}{C^2}) \cdot {\Phi_{c,l}^t}^4 \cdot \widetilde{\text{ReLU}}'(\Phi_{c,l}^t)$$

*Proof.* For any $c \in [C], r \in [m], l \in [2]$, we have from Claim 2'

$$\langle \Delta\boldsymbol{\theta}_{c,r}^t, v_{c,l}\rangle = \eta\mathbb{E}_{\boldsymbol{x}_1,\boldsymbol{x}_2 \sim \hat{\mathcal{D}}^2}\left[\left(\tilde{k}_c^{\mathcal{T}}(\boldsymbol{x}_1, \boldsymbol{x}_2) - \tilde{k}_c(\boldsymbol{x}_1, \boldsymbol{x}_2)\right)^+ \left[\sum_{j \in \{1,2\}} \Psi_{c,r}(\boldsymbol{x}_j)\left(V_{c,r,l}(\boldsymbol{x}_{3-j}) - \tilde{O}(\sigma_p P)\right)\right]\right]$$

Note that as $\mu \leq \frac{1}{\text{poly}(C)}$, we can suppose that both $\boldsymbol{x}_1, \boldsymbol{x}_2$ are multi-view data. Using Claim 2, we have that

$$\langle \Delta\boldsymbol{\theta}_{c,r}^t, v_{c,l}\rangle \geq \eta\mathbb{E}_{\boldsymbol{x}_1,\boldsymbol{x}_2 \sim \hat{\mathcal{D}}^2}\left[\left(\tilde{k}_c^{\mathcal{T}}(\boldsymbol{x}_1, \boldsymbol{x}_2) - \tilde{k}_c(\boldsymbol{x}_1, \boldsymbol{x}_2)\right)^+ \left[\sum_{j \in \{1,2\}} \Psi_{c,r}(\boldsymbol{x}_j)\left(V_{c,r,l}(\boldsymbol{x}_{3-j}) - \tilde{O}(\sigma_p P)\right)\right]\right]$$

Let $r = \text{argmax}_{r' \in [m]}\{\langle\boldsymbol{\theta}_{c,r'}^t, v_{c,l}\rangle\}$, such that definitely $\langle\boldsymbol{\theta}_{c,r}^t, v_{c,l}\rangle \geq \tilde{\Omega}(\Phi_{c,l}^t)$ because $m = \text{polylog}(C)$.

But if $\boldsymbol{x}$ is multi-view and $v_{c,l} \in \mathcal{V}(\boldsymbol{x})$ such that $\sum_{p \in \mathcal{P}_{v_{c,l}}} z_p^4 = \Theta(1)$, and also by **Fact A.a** we have that:

$$V_{c,r,l}(\boldsymbol{x}) \geq \Omega(1) \cdot \widetilde{\text{ReLU}}'\left(\langle\boldsymbol{\theta}_{c,r}^t, v_{c,l}\rangle\right) \geq \tilde{\Omega}\left(\widetilde{\text{ReLU}}'(\Phi_{c,l}^t)\right)$$
$$\Psi_{c,r}(\boldsymbol{x}) \geq \sum_{p \in \mathcal{P}_{v_{c,l}}} \widetilde{\text{ReLU}}(\langle\boldsymbol{\theta}_{c,r}^t, v_{c,l}\rangle z_p - \tilde{o}(\sigma_0)) \geq \tilde{\Omega}({\Phi_{c,l}^t}^4)$$

Now, we have assumed that $(c, 3 - l) \notin \mathcal{M}_{\mathcal{T}}$, such that the teacher model has learnt attribute $(c, l)$ and satisfies $\tilde{k}_c^{\mathcal{T}}(\boldsymbol{x}, \boldsymbol{x}') = 1$ when $v_{c,l} \in \mathcal{V}(\boldsymbol{x}) \cap \mathcal{V}(\boldsymbol{x}')$ (for large enough polylogarithmic $m$).

Also, it is simple to see that when $v_{c,l} \in \mathcal{V}(\boldsymbol{x}) \cap \mathcal{V}(\boldsymbol{x}')$ & $v_{c,3-l} \notin \mathcal{V}(\boldsymbol{x}) \cap \mathcal{V}(\boldsymbol{x}')$, for large enough $m$, that $\Phi_{c,l}^t \leq \frac{1}{2m}$ implies that $k_c(\boldsymbol{x}, \boldsymbol{x}') \leq \frac{1}{m^2}$ by Lemma 1, and so $\tilde{k}_c(\boldsymbol{x}, \boldsymbol{x}') = 0$, i.e. the student has not (yet) learnt $v_{c,l}$.

So we see there are two more conditions that must be satisfied in order for $\left(\tilde{k}_c^{\mathcal{T}}(\boldsymbol{x}_1, \boldsymbol{x}_2) - \tilde{k}_c(\boldsymbol{x}_1, \boldsymbol{x}_2)\right)^+ = 1 > 0$:

1. $v_{c,l} \in \mathcal{V}(\boldsymbol{x}_1) \cap \mathcal{V}(\boldsymbol{x}_2)$ so that $\tilde{k}_c^{\mathcal{T}}(\boldsymbol{x}_1, \boldsymbol{x}_2) = 1$

2. $v_{c,3-l} \notin \mathcal{V}(\boldsymbol{x}_1) \cap \mathcal{V}(\boldsymbol{x}_2)$ so that $\tilde{k}_c(\boldsymbol{x}_1, \boldsymbol{x}_2) = 0$.

Going back to App. B.1.1, we know that these conditions occur with probability $\frac{s^2}{C^2}(1 - o(1))$ for independently sampled $\boldsymbol{x}_1, \boldsymbol{x}_2 \sim \hat{\mathcal{D}}$. Finally, putting everything together we have that:

$$\langle\boldsymbol{\theta}_{c,r}^{t+1}, v_{c,l}\rangle^+ - \langle\boldsymbol{\theta}_{c,r}^t, v_{c,l}\rangle^+ \geq \tilde{\Omega}(\frac{\eta s^2}{C^2}){\Phi_{c,l}^t}^4 \cdot \widetilde{\text{ReLU}}'(\Phi_{c,l}^t)$$

summing over $r' \in [m]$ and noting $\langle \Delta\boldsymbol{\theta}^t_{c,r'}, v_{c,l} \rangle \geq 0 \; \forall r'$, up to small error (as $\sigma_p P = \frac{1}{\text{poly}(C)}$ for a large polynomial), gives us our result.

$\square$

Lemma 2 immediately gives us the following corollaries, because $\Phi^{T_1}_{c,l} \geq \tilde{\Omega}(\sigma_0)$ from Allen-Zhu & Li (2020) Induction Hypothesis F.1.g and $\widetilde{\text{ReLU}}'$ is increasing.

**Corollary 1.** *Define iteration threshold* $T_2 = \tilde{\Theta}(\frac{C^2}{\eta s^2 \sigma_0^7}) = \tilde{\Theta}(\frac{C^{5.1}}{\eta})$, *then for every* $(c,l)$ *such that* $(c, 3-l) \notin \mathcal{M}_{\mathcal{T}}$ *we have:*

$$\Phi^{T_1+T_2}_{c,l} \geq \frac{1}{4m}$$

But likewise, we can also bound the growth of $\Phi_{c,l}$

**Lemma 3.** *If* $(c, 3-l) \in \mathcal{M}_{\mathcal{S}} \backslash \mathcal{M}_{\mathcal{T}}$, *once* $\Phi^t_{c,l} \geq \sqrt{\frac{\log\log(C)}{m}}$, *it no longer gets updated (for large* $C$*):*

*Proof.* Recall Definitions 3 and 4 that $\Phi^*_{c,l} = \sum_{r \in [m]}[\langle \boldsymbol{\theta}^t_{c,r}, v_{c,l} \rangle]^+$ and $\Upsilon^t_{c,l} = \sum_{r \in [m]}[\langle \boldsymbol{\theta}^t_{c,r}, v_{c,l} \rangle]^{+^2}$

Hence by Cauchy-Schwarz we have:

$$\Phi^t_{c,l}{}^2 \leq m\Upsilon^t_{c,l} \implies \Upsilon^t_{c,l} \geq \frac{\log\log(C)}{m^2} \geq \frac{2}{0.4^2 m^2}$$

for large enough $C$. Thus, if $x_1, x_2$ are both multi-view and $v_{c,l} \in \mathcal{V}(\boldsymbol{x}_1) \cap \mathcal{V}(\boldsymbol{x}_2)$, by Lemma 1 we must have

$$k_c(\boldsymbol{x}_1, \boldsymbol{x}_2) \geq \frac{1}{m^2}$$

It's also not difficult to check that any other possible setting for $\boldsymbol{x}_1, \boldsymbol{x}_2$, and $\mathcal{V}(\boldsymbol{x}_1) \cap \mathcal{V}(\boldsymbol{x}_2)$ will lead to $\left(\tilde{k}^{\mathcal{T}}_c(\boldsymbol{x}_1, \boldsymbol{x}_2) - \tilde{k}_c(\boldsymbol{x}_1, \boldsymbol{x}_2)\right)^+ = 0$, and hence

$$\left(\tilde{k}^{\mathcal{T}}_c(\boldsymbol{x}_1, \boldsymbol{x}_2) - \tilde{k}_c(\boldsymbol{x}_1, \boldsymbol{x}_2)\right)^+ = 0 \qquad \forall \boldsymbol{x}_1, \boldsymbol{x}_2$$

$\square$

## C.5 WRAPPING UP PROOF OF THEOREM 2

*Proof.* We are now ready to wrap up our proof. Recall from the proof of Theorem 1 in Allen-Zhu & Li (2020), that after the initial phase of $T_1$ steps of student training on its own:

$$\Phi^{T_1}_c \geq \Omega(\log(C)) \;\; \forall c \in [C],$$

and more specifically:

- If $(c, 3-l) \notin \mathcal{M}_{\mathcal{S}}$, then

$$\Phi^{T_1}_{c,l} \geq \Omega(\log(C))$$

  This gives us perfect test accuracy on the multi-view data, and 50% accuracy on the single-view data, so $0.5\mu$ test accuracy overall without distillation, as per Theorem 1.

- Moreover, we have that if $(c, 3 - l) \notin \mathcal{M}_\mathcal{T}$ and $(c, 3 - l) \in \mathcal{M}_\mathcal{S}$, then by Corollary 1 and Lemma 3:

$$\sqrt{\frac{\log\log(C)}{m}} \geq \Phi_{c,l}^{T_1 + T_2} \geq \frac{1}{4m}$$
$$\implies \sqrt{\log\log(C)} \geq \Phi_{c,l}^{T_1 + T_2} \geq \frac{1}{4m}$$

We see that this change in $\Phi_{c,l}^{T_1 + T_2}$ after FKD training is much smaller than $\Omega(\log(C))$ and so the student after FKD training still has perfect multi-view accuracy, as well as correct predictions on any single-view data that possess the attributes learnt in the initial phase of training.

On the other hand, for single-view data, we know that if we have data point $\boldsymbol{x}, y$ and attribute $v_{c,l} \in \mathcal{V}(\boldsymbol{x})$, such that $c \neq y$, then $\sum_{p \in \mathcal{P}_{v_{c,l}}(\boldsymbol{x})} z_p = \Gamma = O(\frac{1}{\text{polylog}(C)})$, as defined in App. B.1.1.

Hence for small enough $\Gamma$ ($\ll \frac{1}{m}$) we have that if $(c, 3 - l) \notin \mathcal{M}_\mathcal{T}$ and the single view data $\boldsymbol{x}$ is of class $c$ with $\hat{l}(\boldsymbol{x}) = l$ then we have correct prediction, as per Claim 1.

Combining these means that we have correct prediction for any single-view data $\boldsymbol{x}$ of class $c$, and $\hat{l}(\boldsymbol{x}) = l$ such that $(c, 3 - l) \notin \mathcal{M}_\mathcal{T} \cap \mathcal{M}_s$.

By the independence of these sets we have that $|\mathcal{M}_\mathcal{T} \cap \mathcal{M}_s| = (2^{-E})k(1 - o(1))$ this means we have test error less than $(2^{-E-1} + \epsilon)\mu$ for any $\epsilon > 0$, for large enough $C$ as required.

$\square$

# D   DIFFERENCES BETWEEN FKD & OTHER FEATURE KERNEL BASED KD METHODS

In this section, we highlight how our FKD approach overcomes some of the shortcomings of previous feature kernel based KD methods which only use pairwise evaluations of the feature kernel: SP (Tung & Mori, 2019) and RKD (Park et al., 2019). One advantage of FKD relative to these previous works is that we have shown FKD is amenable to ensemble distillation. Moreover, it goes without saying that Feature Regularisation, which arises naturally thanks to our feature kernel learning perspective in Section 4, is already a significant departure that improves FKD relative to SP & RKD. However, even without FR we observe in Section 5 that FKD outperforms both RKD & SP across different datasets and architectures, which warrants explanation.

## D.1   IMPORTANCE OF ZERO DIAGONAL DIFFERENCES: SP (TUNG & MORI, 2019)

First, we consider diagonal kernel differences, $k^\mathcal{S}(\boldsymbol{x}, \boldsymbol{x}) - k^\mathcal{T}(\boldsymbol{x}, \boldsymbol{x})$ for fixed $\boldsymbol{x}$, and motivate using zero diagonal differences, which is not present in SP (Tung & Mori, 2019) but is in FKD thanks to our use of the correlation kernel. Fig. 6 displays this comparison between FKD & SP graphically.

> **Intuition: Downside of non-zero diagonal differences**
>
> The key intuition, which we detail below using our theoretical setup, is that non-zero diagonal differences $k(\boldsymbol{x}, \boldsymbol{x}) - k^\mathcal{T}(\boldsymbol{x}, \boldsymbol{x}) \neq 0$ encourage the student to learn noise in input $\boldsymbol{x}$, compared to when we have zero diagonal differences $k(\boldsymbol{x}, \boldsymbol{x}) - k^\mathcal{T}(\boldsymbol{x}, \boldsymbol{x}) = 0$. In the latter case, we only have non-zero differences for $k(\boldsymbol{x}, \boldsymbol{x}') - k^\mathcal{T}(\boldsymbol{x}, \boldsymbol{x}')$ where $\boldsymbol{x} \neq \boldsymbol{x}'$.

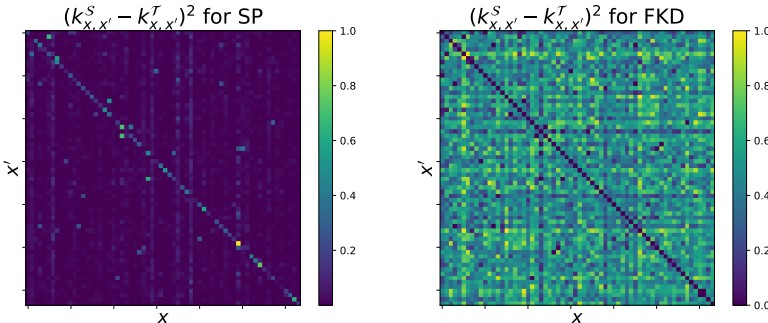

**Figure 6:** Comparison of (normalised) squared differences in $k_{\boldsymbol{x},\boldsymbol{x}'} = k(\boldsymbol{x}, \boldsymbol{x}')$ between student $\mathcal{S}$ & teacher $\mathcal{T}$, across a minibatch of size 64 of CIFAR-100 training data, for SP (left) and FKD (right). We see that whereas FKD has zero diagonal differences, SP is largely dominated by non-zero diagonal differences. Note there is a slight abuse of notation here, in that we plot squared differences in *normalised* kernels, so that FKD uses the correlation kernel and SP uses row-normalisation (Tung & Mori, 2019).

**Diagonal updates** Consider our parameter update Eq. (25) when $\boldsymbol{x}_1 = \boldsymbol{x}_2 = \boldsymbol{x}$:
If we didn't have zero diagonal differences and instead $\tilde{k}_c(\boldsymbol{x}, \boldsymbol{x}) - \tilde{k}_c^{\mathcal{T}}(\boldsymbol{x}, \boldsymbol{x}) = -1$, then:

$$\Delta\boldsymbol{\theta}_{c,r}(\boldsymbol{x}, \boldsymbol{x}) = 2\eta\Psi_{c,r}(\boldsymbol{x})\sum_{p=1}^{P}\widetilde{\text{ReLU}}'(\langle\boldsymbol{\theta}_{c,r}, \boldsymbol{x}_p\rangle)\boldsymbol{x}_p,$$

Now suppose $v = v_{c,1} \in \mathcal{V}(\boldsymbol{x})$. For each $p \in \mathcal{P}_v(\boldsymbol{x})$, recall (from App. B.1.1) that:

$$\boldsymbol{x}_p = z_p v + \xi_p,$$

where we assume zero feature noise for simplicity.
We then see that (c.f. Claim C.13 of Allen-Zhu & Li (2020)):

$$\langle\Delta\boldsymbol{\theta}_{c,r}(\boldsymbol{x}, \boldsymbol{x}), \xi_p\rangle = 2\tilde{\Theta}(\eta)\Psi_{c,r}(\boldsymbol{x})\widetilde{\text{ReLU}}'(\langle\boldsymbol{\theta}_{c,r}, \boldsymbol{x}_p\rangle) + O(\frac{1}{\sqrt{d}}),$$

as $\langle v, \xi_p\rangle = O(\frac{1}{\sqrt{d}})$ with high probability.
But at the same time (by e.g. Claim F.6 of Allen-Zhu & Li (2020)):

$$\langle\Delta\boldsymbol{\theta}_{c,r}(\boldsymbol{x}, \boldsymbol{x}), v\rangle = 2\eta\Psi_{c,r}(\boldsymbol{x})V_{c,r,1}(\boldsymbol{x})(1 \pm o(1))$$

and note that that $V_{c,r,1}(\boldsymbol{x}) \leq \Theta(1)$ by definition. Moreover, in order for the student network to learn attribute $v_{c,1}$, then eventually it must satisfy $\max_{r'}\langle\boldsymbol{\theta}_{c,r'}, v\rangle = \tilde{O}(1)$ from Allen-Zhu & Li (2020). Thus, for $\varrho$ small enough, if $r = \text{argmax}_{r'}\langle\boldsymbol{\theta}_{c,r'}, v\rangle$, we have $\widetilde{\text{ReLU}}'(\langle\boldsymbol{\theta}_{c,r}, \boldsymbol{x}_p\rangle) = 1$.

**Non-diagonal updates** On the other hand, if $\boldsymbol{x}_1 \neq \boldsymbol{x}_2$ and we have $v \in \mathcal{V}(\boldsymbol{x}_1) \cap \mathcal{V}(\boldsymbol{x}_2)$:

$$\Delta\boldsymbol{\theta}_{c,r}(\boldsymbol{x}_1, \boldsymbol{x}_2) = \eta\sum_{j=1,2}\Psi_{c,r}(\boldsymbol{x}_j)\sum_{p=1}^{P}\widetilde{\text{ReLU}}'(\langle\boldsymbol{\theta}_{c,r}, \boldsymbol{x}_{3-j,p}\rangle)\boldsymbol{x}_{3-j,p},$$

and so if $\boldsymbol{x}_1 = \boldsymbol{x}$ and $\xi_p$ denotes the random noise in $\boldsymbol{x}_{1,p}$, then:

$$\langle\Delta\boldsymbol{\theta}_{c,r}(\boldsymbol{x}_1, \boldsymbol{x}_2), \xi_p\rangle = \tilde{\Theta}(\eta)\Psi_{c,r}(\boldsymbol{x}_2)\widetilde{\text{ReLU}}'(\langle\boldsymbol{\theta}_{c,r}, \boldsymbol{x}_{1,p}\rangle) + O(\frac{1}{\sqrt{d}}),$$

as $\langle \boldsymbol{x}_{2,p}, \xi_p \rangle = O(\frac{1}{\sqrt{d}})$. But this time we have

$$\langle \Delta\boldsymbol{\theta}_{c,r}(\boldsymbol{x}_1, \boldsymbol{x}_2), v \rangle = \eta\big(\Psi_{c,r}(\boldsymbol{x}_1)V_{c,r,1}(\boldsymbol{x}_2) + \Psi_{c,r}(\boldsymbol{x}_2)V_{c,r,1}(\boldsymbol{x}_1)\big)(1 \pm o(1)),$$

We see that whereas in diagonal updates $\Delta\boldsymbol{\theta}_{c,r}(\boldsymbol{x}, \boldsymbol{x})$ the increments for $\langle \Delta\boldsymbol{\theta}_{c,r}(\boldsymbol{x}, \boldsymbol{x}), \xi_p \rangle$ and $\langle \Delta\boldsymbol{\theta}_{c,r}(\boldsymbol{x}, \boldsymbol{x}), v \rangle$ are 1:1, for non-diagonal updates $\Delta\boldsymbol{\theta}_{c,r}(\boldsymbol{x}_1, \boldsymbol{x}_2)$ they are 1:2 respectively.

Thus, the parameter updates for $\boldsymbol{\theta}_{c,r}$ with non-zero diagonal differences in feature kernels are more likely to learn noise, $\xi_p$, compared to our zero diagonal updates which rely only on non-diagonal $\Delta\boldsymbol{\theta}_{c,r}(\boldsymbol{x}_1, \boldsymbol{x}_2)$ for $\boldsymbol{x}_1 \neq \boldsymbol{x}_2$. This is why we zero out diagonal differences for FKD, using the feature correlation matrix in practice.

### D.2 PROBLEM OF HOMOGENEOUS NNS IN RKD (PARK ET AL., 2019)

The *distance-wise* version of RKD (Park et al., 2019) is as follows: for $\boldsymbol{x}, \boldsymbol{x}'$, we calculate $\psi_{\mathcal{T}}(\boldsymbol{x}, \boldsymbol{x}') = \|h_{\mathcal{T}}(\boldsymbol{x}, \boldsymbol{\theta}_{\mathcal{T}}) - h_{\mathcal{T}}(\boldsymbol{x}', \boldsymbol{\theta}_{\mathcal{T}})\|_2$, where recall $h_{\mathcal{T}}$ is the last-layer teacher feature extractor. Likewise, we also calculate $\psi_{\mathcal{S}}(\boldsymbol{x}, \boldsymbol{x}') = \|h_{\mathcal{S}}(\boldsymbol{x}, \boldsymbol{\theta}_{\mathcal{S}}) - h_{\mathcal{S}}(\boldsymbol{x}', \boldsymbol{\theta}_{\mathcal{S}})\|_2$. The RKD loss adds $\lambda_{\mathrm{KD}} \mathbb{E}_{\boldsymbol{x}, \boldsymbol{x}'}[(\psi_{\mathcal{T}}(\boldsymbol{x}, \boldsymbol{x}') - \psi_{\mathcal{S}}(\boldsymbol{x}, \boldsymbol{x}'))^2]$ to the student's training loss.[9] While RKD (Park et al., 2019) does ensure zero diagonal differences, i.e. that $\psi_{\mathcal{T}}(\boldsymbol{x}, \boldsymbol{x}) - \psi_{\mathcal{S}}(\boldsymbol{x}, \boldsymbol{x}) = 0$, it suffers from a related issue, due to the homogeneity of NNs that use ReLU nonlinearity, which is ubiquitous in image classification tasks.

Suppose we take $\boldsymbol{x}$ and define $\boldsymbol{x}' = M\boldsymbol{x}$ for some $M > 0$. For example, think of taking a cat image and multiplying all the pixel values by $M$. For ReLU (C)NNs without bias parameters, we have that $h(\boldsymbol{x}, \boldsymbol{\theta})$ is 1-homogeneous: $h(\boldsymbol{x}', \boldsymbol{\theta}) = Mh(\boldsymbol{x}, \boldsymbol{\theta})$. This means that it is likely (depending on the norms of the features $h_{\mathcal{S}}$ and $h_{\mathcal{T}}$) that we will have $\psi_{\mathcal{T}}(\boldsymbol{x}, M\boldsymbol{x}) - \psi_{\mathcal{S}}(\boldsymbol{x}, M\boldsymbol{x}) \neq 0$. But a cat image multiplied by some scalar $M$ is still a cat image, hence RKD runs into the same problems as in App. D.1 of learning noise in $\boldsymbol{x}$.

On the other hand for FKD: correlation kernel $\rho_{\mathcal{S}}(\boldsymbol{x}, M\boldsymbol{x}) = \rho_{\mathcal{T}}(\boldsymbol{x}, M\boldsymbol{x}) = 1$, hence $\rho_{\mathcal{S}}(\boldsymbol{x}, M\boldsymbol{x}) - \rho_{\mathcal{T}}(\boldsymbol{x}, M\boldsymbol{x}) = 0 \quad \forall \boldsymbol{x} \in \mathbb{R}^d, M > 0$.

## E  PYTORCH-STYLE PSEUDOCODE FOR FKD

In Alg. 2, we provide PyTorch-style Paszke et al. (2019b) pseudocode for the distillation and feature regularisation losses in FKD. We note that FKD only requires pairwise computations of feature (correlations) kernels. This alleviates the need for matrix multiplication/inversion operations with batch-by-batch size matrices, which is beneficial for scalability.

## F  EXPERIMENTAL DETAILS AND FURTHER RESULTS

### F.1  FIG. 2: PREDICTIVE DISAGREEMENT ACROSS INDEPENDENT INITIALISATIONS VS RETRAINED LAST LAYER

All models are ResNet20v1 trained with standard hyperparameters:

- 160 epochs training time with batch size 128 and learning rate 0.1 which is decayed by a factor of 10 after epochs 80 and 120.
- SGD optimiser with momentum 0.9 and weight decay of 0.0001.

---

[9]We do not consider the angle-wise RKD loss here, but there will be similar issues due to homogeneity.

---

**Algorithm 2** PyTorch-style pseudocode for Feature Kernel Distillation (FKD).

---

```
# B:      Batch size.
# L_FKD: FKD regularisation strength.
# L_FR:  Feature regularisation strength.
# D_s:    Student feature dimension.
# D_t:    Teacher feature dimension.

# f_s:    Student features B x D_s
# f_t:    Teacher features B x D_t

# mm: matrix-matrix multiplication

# Compute student feature correlation kernel matrix s_c
s_k = mm(f_s, f_s.T)                                 # B x B
s_k_diag_inv_sqrt = torch.diag(s_k).pow(-1/2)
s_k_diag_inv_sqrt = s_k_diag_inv_sqrt.reshape(-1, 1)  # B x 1
s_c = s_k_diag_inv_sqrt * s_k * s_k_diag_inv_sqrt.T   # B x B

# Compute teacher feature correlation kernel matrix t_c
with torch.no_grad():
    t_k = mm(f_t, f_t.T)                             # B x B
    t_k_diag_inv_sqrt = torch.diag(t_k).pow(-1/2)
    t_k_diag_inv_sqrt = t_k_diag_inv_sqrt.reshape(-1, 1) # B x 1
    t_c = t_k_diag_inv_sqrt * t_k * t_k_diag_inv_sqrt.T  # B x B

distil_loss = ((t_c - s_c).pow(2)).mean()
feat_reg_loss = (f_s.pow(2)).mean()

# FKD loss to be added to supervised loss
loss_fkd = L_FKD * distil_loss + L_FR * feat_reg_loss
```

---

- CIFAR10 data is normalised in each channel such that the training data is zero mean and unit standard deviation. Random crops and horizontal flips used as data augmentation.

- All models are initialised with Kaiming initialisation He et al. (2015).

Out of 10000 test points, the predictive disagreements between a 'reference' model and either: independent initialisations (top row) or retrained last layers (bottom row) are depicted in Table 4. We see two clear trends. First, the retrained last layer has much fewer disagreements with the reference model than an independent initialisation model, highlighting the importance of the feature kernel. Secondly, the vast majority of disagreements between independent initialisations are where one of the models is correct. This reinforces our intuition/theoretical analysis that ensembling NN works because different initialisations bias the models to capture different useful features, and hence ensemble distillation (via feature kernels) can improve student performance.

**Table 4:** Breakdown of predictive disagreements between reference and alternate models over 10000 CIFAR10 test points, in terms of which model (if any) was correct. Mean $\pm$ standard deviations over 3 independent initialisations for top row, and over 3 independent reference models for bottom row. All models achieved between 8.0%-8.5% test error.

| Alternate model | Reference correct | Alternate correct | Neither correct | Total disagreement |
|---|---|---|---|---|
| Independent Init | $350 \pm 14.6$ | $383 \pm 15.9$ | $124 \pm 3.7$ | $857 \pm 29.8$ |
| Retrained LL | $30 \pm 2.9$ | $35 \pm 1.7$ | $15 \pm 5.4$ | $80 \pm 4.1$ |

## F.2   FIG. 3: ADDITIONAL FEATURE KERNEL HISTOGRAMS

In Fig. 7, we provide additional plots to Fig. 3 that depict the difference in distribution (over $x$) of feature kernel values $k(x, x)$, for FKD with and without Feature Regularisation (FR).

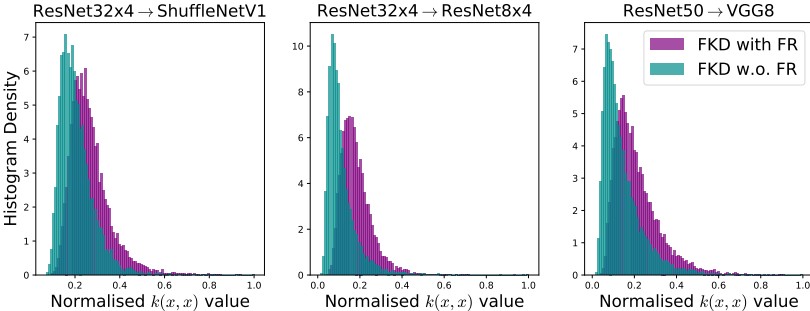

**Figure 7:** Comparison of normalised $k(\boldsymbol{x}, \boldsymbol{x})$ values between FKD with & without Feature Regularisation (FR), across different Teacher→Student architectures, on CIFAR-100 test set. We see, like in Fig. 3 that FR encourages a more even distribution of $k(\boldsymbol{x}, \boldsymbol{x})$ across $\boldsymbol{x}$, for all architectures.

**Negative hypothesis.** We originally hypothesised that FR could benefit FKD, in addition to balancing the distribution of $k(\boldsymbol{x}, \boldsymbol{x})$, as it could reduce the sparsity in the NN last-layer representation activations, which is consistent with the proofs of Theorems 1 and 2. Indeed, the NN predictions are dominated by a select few neurons who 'have won the lottery' (Allen-Zhu & Li, 2020) on account of being most correlated with one of the attributes $\{v_{c;l}\}_{l \in [2], c \in [C]}$ at random initialisation. In our analysis, as few as $O(\log^5(m))$ out of $m$ neurons could be inactive. From our feature kernel learning perspective, this seems like a highly undesirable phenomenon, because if $k(\boldsymbol{x}, \boldsymbol{x}') = \langle h(\boldsymbol{x}; \boldsymbol{\theta}), h(\boldsymbol{x}'; \boldsymbol{\theta}) \rangle = \sum_{r=1}^{Cm} h_r(\boldsymbol{x}, \boldsymbol{\theta}) h_r(\boldsymbol{x}', \boldsymbol{\theta})$ only has useful contributions from a dominant minority of $r \in [Cm]$, then we are not utilising the full capacity of the model. FR seemed appropriate to reduce this sparsity as $\ell_2$ regularisation is known to promote non-sparse solutions (Van Den Doel et al., 2013). However, we experimentally found the opposite to our hypothesis: that FR trained FKD student had more inactive neurons relative to FKD students trained without FR. This again highlights that, while (we believe) our results in this work provide compelling evidence to highlight the validity of feature kernel based distillation, there are still gaps between our theory and practice, and further questions to be answered in future work.

### F.3  FIG. 4: ENSEMBLE DISTILLATION

All individual VGG8 networks that made up the teacher ensemble were trained using the default training regime from Tian et al. (2020), with independent parameter initialisations. Indeed, all of our experiments in Section 5 used Tian et al. (2020)'s excellent open-source PyTorch codebase (Paszke et al., 2019a).[10]

For student networks, we used the training regime for vanilla KD from Tian et al. (2020) for all ensemble sizes. For FKD, we used the hyperparameters from our ResNet50→VGG8 experiment in Table 2 for all ensemble sizes.

### F.4  TABLE 1: DATASET TRANSFER

The VGG13 teacher checkpoint is provided by Tian et al. (2020). For both CIFAR-10 and STL-10, all student networks are trained for 160 epochs with batch size 64 using SGD with momentum, with learning rate decays at epochs 80, 120, 150. The student trained without KD used default hyperparameters from Tian et al. (2020), which are indeed strong hyperparameters for standard training. For FKD, RKD (Park et al., 2019) and SP (Tung & Mori, 2019), we tuned the learning

---

[10]https://github.com/HobbitLong/RepDistiller

rate, learning rate decay, and KD regularisation strength $\lambda_{KD}$ on a labeled validation set of size 5000 for CIFAR-10 and 1000 for STL-10, before retraining using best hyperparameters on the full training(+unlabeled) dataset. We also tuned the FR regularisation strength, $\lambda_{FR}$ for FKD when FR was used. All RKD and SP hyperparameters were tuned in a large window around their default values from Tian et al. (2020), which were all author recommended. For FKD, we allowed $\lambda_{KD}$ to range in [1,1000], and $\lambda_{FR}$ to range in [0,20]. All hyperparameters sweeps were conducted using Bayes search.

For STL-10, we used a batch size of 512 for all KD methods' regularisation terms, compared to 64 for the standard cross-entropy loss. This was due to the fact that STL-10 has only 5K labeled datapoints, and we wanted to ensure that the student used as much of the unlabeled data as possible for each feature-kernel based KD method's additional regularisation term during 160 epochs of training. 512 batch size was the maximum power of 2 before we ran into memory issues on a 11GB VRAM GPU, which occured for the RKD method.

Both CIFAR-10 and STL-10 data are normalised in each channel such that the training data is zero mean and unit standard deviation. Random crops and horizontal flips used as data augmentation. STL-10 images are downsized from 96x96 to 32x32 resolution.

### F.5    TABLE 2: CIFAR-100 AND IMAGENET COMPARISON

**CIFAR-100.** All networks were trained for 240 epochs with batch size 64, with learning rate decay at epochs 150, 180, 210 using SGD with momentum. All teacher networks use the exact same checkpoints as provided by Tian et al. (2020). Learning rate, learning rate decay, $\lambda_{KD}$, and $\lambda_{FR}$ (when used) were tuned as in App. F.4 on a validation of size 5000. All other hyperparameters were set to the default values used by Tian et al. (2020). The CIFAR-100 data is normalised in each channel such that the training data is zero mean and unit standard deviation, with random crops and horizontal flips used for data augmentation. All results provided denote the test set accuracy at the end of the 240 epochs of training.

**ImageNet.** The ImageNet dataset (ILSVRC-2012) consists of about 1.3 million training images and 50,000 validation images from 1,000 classes. Each training image is extracted as a randomly sampled 224x224 crop or its horizontal flip without any padding operation. All teacher networks use the exact same checkpoints as provided by Chen et al. (2021). The initial learning rate is 0.1 and divided by 10 at 30 and 60 of the total 90 training epochs. We set the mini-batch size to 256 and the weight decay to $10^{-4}$. $\lambda_{KD}$, and $\lambda_{FR}$ (when used) were tuned as in App. F.4 on a validation of size 5000 except using Bayes search. All results are reported in a single trial. All other hyperparameters were set to the default values used by Chen et al. (2021). All results provided denote the Top-1 test accuracy (%). Accuracy of baselines were reported in Tian et al. (2020).

### F.6    SENSITIVITY TO $\lambda_{KD}$

In Fig. 8, we plot the sensitivity of FKD to the strength of the distillation regularisation $\lambda_{KD}$ in Eq. (2) for the VGG13→VGG8 experiment on CIFAR-100. We see that a well tuned $\lambda_{KD}(\approx 300$ here) is important for best student generalisation. Feature regularisation $\lambda_{FR} = 20$ in Fig. 8.

### F.7    TABLE 5: ANALYSES IN NEURAL MACHINE TRANSLATION

In this section, We performed analyses for a neural machine translation (NMT) task proposed by Tan et al. (2019). In the analyses, we could only obtain data for En-De (from English to German) translation since links to the datasets for other languages are broken. Therefore, we employed a self-distillation method on a pre-trained English model for En-De translation as follows:

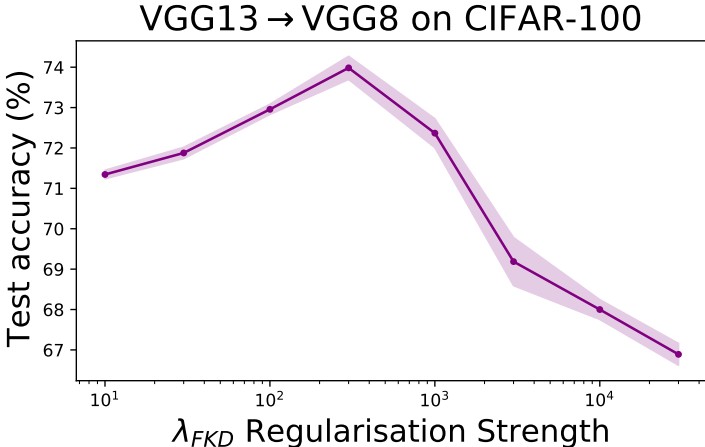

**Figure 8:** Comparison of normalised $k(\boldsymbol{x}, \boldsymbol{x})$ values between FKD with & without Feature Regularisation (FR), across different Teacher→Student architectures, on CIFAR-100 test set. We see, like in Fig. 3 that FR encourages a more even distribution of $k(\boldsymbol{x}, \boldsymbol{x})$ across $\boldsymbol{x}$, for all architectures.

- We train a single teacher transformer model on the IWSLT dataset for English (Tan et al., 2019).
- We perform self-distillation on the teacher model for En-De translation (Tan et al., 2019).
- We did not search for optimal hyperparameters, and used default parameters of the code provided by the authors of Tan et al. (2019). The results are given in Table 5.

**Table 5:** BLEU of the teacher model of (Tan et al., 2019) (Teacher), self-distillation of (Tan et al., 2019) (SD), SD with KD of (Hinton et al., 2015), SD with FKD, and SD with FKD loss obtained by replacing distillation loss (2) of (Tan et al., 2019) with FKD, in En - De neural machine translation tasks.

| Teacher (Tan et al., 2019) | SD (Tan et al., 2019) | SD with KD | SD with FKD | SD with FKD loss |
|---|---|---|---|---|
| 27.32 | 27.49 | 27.51 | **27.64** | **27.79** |

We first note that, we adapted vanilla KD and our FKD for sequential data in the KD loss (2) of (Tan et al., 2019) in this task. More precisely, we first computed vanilla KD and FKD on token probabilities, and added these loss functions to the KD loss (eq 2 of (Tan et al., 2019)) in KD and FKD. In the results, aggregating vanilla KD with the KD loss (eq 2 of (Tan et al., 2019)) improved accuracy from 27.49 to 27.51. However, FKD further boosted BLEU to 27.64. We then replaced KD loss (Eq. 2 of (Tan et al., 2019)) with FKD for training. Remarkably, FKD further boosted the BLEU to 27.79. These results suggest that the proposed FKD can be applied in NMT tasks, successfully. We hope that these results will motivate researchers to employ FKD in various different NLP tasks including but not limited to multilingual NMT, named entity recognition and question answering.

### F.8  TABLE 6: ANALYSES IN AUTOMATIC SPEECH RECOGNITION

In this section, we used a CRDNN model (VGG + LSTM,GRU,LiGRU+ DNN) on the TIMIT dataset. In this experiment, we used a distillation approach proposed by Gao et al. (2020) for ASR tasks as follows:

- We train a single teacher model on the TIMIT dataset Ravanelli et al. (2021).

- We perform self-distillation on the teacher Gao et al. (2020).

- We did not search for optimal hyperparameters, and used default parameters of the Speech-Brain Library.

- In this task, replacing CTC/NLL distillation losses with KD (Hinton et al., 2015) did not converge. Additional investigation with hyperparameter search is needed. We used phoneme error rate (PER) to measure accuracy of models.

**Table 6:** Phoneme error rate (PER) of methods in automatic speech recognition tasks.

| Teacher | Distilled Teacher (Gao et al., 2020) | KD (Hinton et al., 2015) | FKD |
|---------|--------------------------------------|--------------------------|--------|
| 13.26   | 12.80                                | 12.86                    | **12.59** |

The results are given in Table 6. Similar to the NMT task, we adapted vanilla KD and our FKD for sequential data as follows: We first computed vanilla KD and FKD loss functions on token probabilities, and then added to the total loss (eq 7 of Gao et al. (2020)). In the analyses, Vanilla KD (Hinton et al., 2015) increased the PER from 12.80 to 12.86. However, FKD further improved the PER from 12.80 to 12.59. In this task, training models by replacing CTC/NLL distillation losses (eq 4 or 5 of Gao et al. (2020)) with KD (Hinton et al., 2015) and FKD did not converge. In conclusion, these results propound that FKD can be applied for different tasks, i.e., image classification, NMT and ASR, boosting accuracy of baseline distillation methods. We hope that these initial results will motivate researchers in different communities (computer vision, NLP, and ASR) to further expound and apply FKD in additional sub-tasks.

