# OpenReview forum: "Feature Kernel Distillation"
_ICLR.cc/2022/Conference — ICLR 2022 Poster_

### Official Review · Reviewer_h8ud · 2021-10-22

**Correctness:** 3
**Technical Novelty And Significance:** 2
**Empirical Novelty And Significance:** 2
**Recommendation:** 6
**Confidence:** 4

**Main Review:**

Authors nicely motivate their work and draw connections with recent theoretical findings. However, I have several concerns regarding this paper:

1) Authors claim that "We further use our theory to motivate practical considerations for improving student generalisation when using distillation with feature kernels, which allows us to propose a novel approach: Feature Kernel Distillation (FKD)." In other words, this work does not only provide a theoretical discussion on feature kernel distillation, but also it is claimed that a novel feature distillation method is proposed. However, there is a vast amount of works that employ similar kernels (even if not explicitly drawing connections with kernels) and employ divergence metrics between the kernels for performing distillation. More specifically, the angle-wise loss used in RKD as proposed in [1] is probably exactly the same (or very similar ) as the formulation used in (3), PKT [2] employs a KL divergence loss based on similar cosine-based kernels (ref to Section 3.1 on correlation kernel, which seems like a cosine-based formulation), while even the work in [3] is highly related, since they proposed MDS-based distillation.

Only [1] is briefly discussed and included in the experimental setup. However, I think authors should very carefully discuss similarities among the proposed method and the related ones.

[1] Park, Wonpyo, et al. "Relational knowledge distillation." Proceedings of the IEEE/CVF Conference on Computer Vision and Pattern Recognition. 2019.

[2] Passalis, Nikolaos, and Anastasios Tefas. "Learning deep representations with probabilistic knowledge transfer." Proceedings of the European Conference on Computer Vision (ECCV). 2018.

[3] Yu, Lu, et al. "Learning metrics from teachers: Compact networks for image embedding." Proceedings of the IEEE/CVF Conference on Computer Vision and Pattern Recognition. 2019.

2) Furthermore, authors seem to also ignore a very large portion of the recent literature on ensemble-based online distillation methods:

[4] Zhang, Ying, et al. "Deep mutual learning." Proceedings of the IEEE Conference on Computer Vision and Pattern Recognition. 2018.
Lan, Xu, Xiatian Zhu, and Shaogang Gong. "Knowledge distillation by on-the-fly native ensemble.", NIPS2018

[5] Chen, Defang, et al. "Online knowledge distillation with diverse peers." Proceedings of the AAAI Conference on Artificial Intelligence. Vol. 34. No. 04. 2020.

Authors heavily discuss topics related to ensemble distillation, so I think that recent works on the topics should be discussed and - most importantly - included in the experimental evaluation.

3) On the theoretical side, I am not sure that similarities can indeed provide useful knowledge in datasets with many classes. For example, would the authors expect their method to scale when 1000 classes are used? What kind of kernel should be employed in this case? In my experience, such methods tend to degenerate when the number of the classes go up, since the discriminative power of most kernels goes down. Therefore authors should better define the application area of the proposed method and/or provide relevant experiments.

4) It is unclear if the proposed method is expected to work when activation functions other than ReLU are used.

5) Is the proposed method only applicable when performing distillation from the last fully connected layer?

**Summary Of The Paper:**

In this paper the authors consider neural networks as data-dependent kernel machines and propose applying a distillation method directly on the pairwise kernel matrix of the models. Authors extend their setting into ensemble settings, examining both some theoretical aspects of the process, building upon the work of Allen-Zhu & Li (2020), as well experimentally. More specifically, authors provide experiments on ensemble distillation on CIFAR-100, as well as on dataset knowledge transfer on CIFAR-100 to STL-10 and CIFAR-10.

**Summary Of The Review:**

Overall, this is a well written paper, easy to follow and the proposed method is - mostly - sound and backed up with some theoretical evidence. However, given the amount of literature ignored, together with the high similarity of the proposed method with many existing ones, I am very skeptical of this paper at this current form. Also, authors should provide more in depth experiments to evaluate how the employed kernel behaves when more classes are used, as well as when there are architecture changes. In other works, the scope and application area of the method should be clearly defined and possible limitation clearly discussed. As a minor point, I feel that some claims are quite bold, e.g., "This enables us to *prove* that KD using only pairwise feature kernel comparisons can improve NN performance in such settings".

---

> ### Author Response · Authors · 2021-11-15
> **Thank you! Initial author response below. [1/n] Related work**
>
> Thank you for taking the time to review our paper and for your constructive feedback. Please find replies to your concerns below:
>
> **1) Related work: *"...I  think  authors  should very  carefully  discuss  similarities  among  the  proposed  method  and  the  related  ones."***
>
> Thank you for informing us of these related works.  We were only aware of RKD [1] (which we have discussed in our paper) and PKT [2], and will be sure to cite [2]-[5] in our related works section.
>
> We respectfully disagree with the reviewer’s point that we have not sufficiently compared  to  related  works.   We  agree  that  there  are  similar  methods  to  FKD  in  the existing literature, but we believe we have been clear and fair regarding the distinction  between  FKD  and  the  most  related  works:  Relational  Knowledge Distillation (RKD) [1] and Similarity Preserving (SP) (Tung & Mori 2019).  We disagree we have only ‘briefly discussed’ RKD, as we have dedicated an entire section of our Appendix (section D) to compare FKD to these most related works, SP and RKD. In Appendix D we  highlight  that  both  SP  and  RKD  are  more  likely  to  learn  noise  compared  to FKD  (due  to  methodological  differences)  in  our  theoretical  setup,  which  also  helps explain  why  FKD  *without*  feature  regularisation  outperforms  both  RKD  and  SP throughout our experiments.  It goes without saying that feature regularisation is a novel  methodological  aspect  of  FKD  which  is  not  present  (as  far  as  we  are  aware) in any previous related work but provides significant performance improvements (c.f.Table 1 and Table 2), and highlights the benefit of our approach of focusing on how best to make the student’s feature kernel amenable to improved generalisation using distillation, rather than previous works which focused purely on extracting knowledge from the teacher’s feature kernel.
>
> We also believed that PKT [2] was a more significant departure from FKD, compared to  RKD  and  SP,  due  to  PKT’s  probabilistic  framing  of  distillation  (which  leads  to a different methodology, and moreover a potentially troublesome dependence on the batch size/ dependence between batch elements due to the normalisation to obtain probabilities, which introduces dependence  across  batch  samples,  unlike  FKD  which  purely  works  with  pairwise  feature kernel evaluations over the batch), and hence we did not compare to PKT in our experimental setup.  However, in light of the reviewer’s comments we have rerun (with equal hyperparameter tuning) PKT on our dataset transfer experiments (Table 1) where we see that it is outperformed by FKD (apart from on the easier CIFAR10 task where it matches FKD up to statistical significance). We have additionally added a CIFAR100$\rightarrow$Tiny-ImageNet experiment. Test accuracies (mean+95% confidence) shown below:
>
> - CIFAR100$\rightarrow$CIFAR10: PKT $92.58\pm0.06$ vs FKD $92.61\pm 0.07$
> - CIFAR100$\rightarrow$STL10: PKT $74.18\pm0.18$ vs FKD $75.44\pm 0.31$
> - CIFAR100$\rightarrow$Tiny-ImageNet: PKT $49.29\pm0.09$ vs FKD $50.67\pm 0.12$
>
> The other works you put forward, [3]-[5], we do not believe are as closely related to FKD. [3] because it requires labels to define their Triplet Loss, Eq. 1 in [3], as opposed to FKD (and indeed SP, RKD & PKT) which all work without labels.  As for [4]-[5], online ensemble distillation is a very interesting area, out of the scope of this work, which  could  potentially  serve  as  future  work  for  FKD.  Moreover,  all  such methods [4]-[5] require all networks (student+teacher) to be trained on the same dataset (withsame classes), which as we stress in Table 1 is not a constraint that FKD suffers from.
>
> On a more general note, we believe that our work is complementary, not competing, with  all  the  aforementioned  works  which  use  a  measure  of  ‘similarity’  or  ‘relation’ between  inputs  to  distill  information.   We  believe  it  is  important  in  deep/machine learning to have theoretically principled methods, and our work provides this for SP, RKD, and to a lesser extent PKT (it is not clear that their loss function is amenable to our analysis).  Methodologically, FKD uses our theoretical insights to derive what we believe is an improved version of these related methods, which we verify experimentally (Table 1 and 2) and theoretically (Appendix D).

---

> > ### Author Response · Authors · 2021-11-15
> > **[2/n] Clarifications and ImageNet experiment**
> >
> > **2) Experiments/Theory: *"Is the proposed method only applicable when performing distillation from the last fully connected layer?***
> >
> > Yes at the moment, as the network is not linear in the earlier layers hence there is no connection to kernels.  Having said that, it may be possible to use the Neural Tangent Kernel (Jacot et al.  2018) to derive settings where the NN can be thought of as linear in the earlier layer parameters.  However, there are known limitations to the NTK analysis, primarily that it forbids feature learning (which is cental to our analyses) due to restrictive assumptions on learning rate/parameterisation, so one would require further advances in NTK theory before allowing FKD to have theoretical justification for pre-final layer feature kernel distillation.  Note that there is nothing stopping one from applying FKD to earlier layers, just that we don’t have theoretical justification to do so, so we did not try.  We also point out that (the related) Similarity Preserving KD (SP) (Tung & Mori 2019) noted that they did not observe any improvement inperformance we using earlier layers of a Wide ResNet for their distillation method.
> >
> > **3) More classes/ImageNet experiment: *would  the  authors  expect  their  method  to  scale  when  1000 classes  are  used?  What  kind  of  kernel  should  be  employed  in  this  case?  In  my  experience,  such  methods  tend  to  degenerate  when  the  number  of  the  classes  go  up,  since the discriminative power of most kernels goes down.?"***
> >
> > Yes we do expect our methods to  scale  to  large  numbers  of  classes  as  our  theoretical  results  specifically  show  the benefits of distillation in the large class regime (cf Theorem 2).
> >
> > To empirically examine  FKD  when  1000  classes  are  used,  we  performed  analyses  on  the ImageNet dataset.  We computed ImageNet test accuracies (%) to compare FKD with KD baselines for the mean over 3 students.  * denotes results from [Inet-5], and we used their provided code for the experiments (with Teacher ResNet34 and Student ResNet18):
> > - Teacher*:  $73.26$
> > - Student*:  $69.67$
> > - KD*:  $70.62$
> > - SP [Inet-4]: $69.99$
> > - [Inet-1]*:  $70.30$
> > - [Inet-2]*:  $68.86$
> > - [Inet-3]*:  $70.37$
> > - FKD: $70.84$
> >
> > (with Teacher ResNet34 and Student ShuffleV2x0.5):
> > - Teacher*:  $73.54$
> > - Student*:  $53.78$
> > - KD*:  $53.73$
> > - SP [Inet-4]: $51.73$
> > - [Inet-1]*:  $ 53.97$
> > - [Inet-2]*:  $51.60$
> > - [Inet-3]*:  $52.96$
> > - FKD: $54.01$
> >
> > The results show that FKD outperforms all the KD baselines for image classification with 1000 classes.
> >
> > In addition, we have run the same setup for Table 1 on the CIFAR-100 to Tiny ImageNet (which has 200 classes) dataset transfer experiment, and we still observe and improvement in performance from FKD, test accuracies as shown below:
> > - Student: $48.53$
> > - SP: $48.73\pm0.14$
> > - RKD: $48.74\pm0.17$
> > - FKD w/o FR: $48.85\pm0.09$
> > - FKD: $50.67\pm0.12$
> >
> > We note that there is no choice of kernel in our setup:  we view the trained NN as a data dependent kernel machine with kernel that is defined (and determined) by the feature kernel (cf Appendix A).
> >
> > One thing to note is that from our theory, with more classes we need a wider NN (ie the last layer feature dimension has to be larger), which also intuitively makes sense (e.g.  ResNet50, with feature dimension 2048, is commonly used for ImageNet).  This is  the  reason  why  we  chose  to  use  VGG8  for  Table  1  (with  feature  dimension  512) rather than e.g.  ResNet20 (with feature dimension 64).  Perhaps this could explain some of the reviewer’s experience?
> >
> > [Inet-1]  S.  Ahn,  S.  X.  Hu,  A.  C.  Damianou,  N.  D.  Lawrence,  and  Z.  Dai,  “Variational  information  distillation  for  knowledge  transfer,”  in  Proceedings  of  the  IEEE Conference on Computer Vision and Pattern Recognition, 2019, pp.  9163–9171.
> >
> > [Inet-2] N. Passalis, M. Tzelepi, and A. Tefas, “Heterogeneous knowledge distillationusing information flow modeling,” in Proceedings of the IEEE Conference on ComputerVision and Pattern Recognition, 2020, pp.  2336–2345.
> >
> > [Inet-3] K. Yue, J. Deng, and F. Zhou, “Matching guided distillation”, ECCV, 2020.
> >
> > [Inet-4]  Tung,  F.  and  Mori,  G.,  ”Similarity-preserving  knowledge  distillation”,  In Proceedings of the IEEE/CVF International Conference on Computer Vision, pages 1365–1374, 2019.
> >
> > [Inet-5]  Chen  et  al.,  Cross-Layer  Distillation  with  Semantic  Calibration,  AAAI’21 (Extended version:  https://arxiv.org/abs/2012.03236).

---

> > > ### Author Response · Authors · 2021-11-15
> > > **[3/3] Clarifications and overstated claims**
> > >
> > > **4) Activation function: *"It is unclear if the proposed method is expected to work when activation functions other than ReLU are used?"***
> > > It would be interesting to consider if our theory can be extended to non ReLU settings, though there is also nothing stopping one from trying in practice.  We would be interested if the reviewer has references for related work on feature-kernel based KD that consider non-ReLU activations (either experimentally or theoretically)?  Certainly, all papers we are aware of ubiquitously use ReLU.
> > >
> > > **4) Architectural changes: *"Authors should provide more in depth experiments to evaluate...  when there are architecture changes"***
> > > We have provided in Table 2 multiple experiments with significant architectural changes between teacher and student:  from 1) ResNet32x4 to ShuffleNetV1, and 2) from ResNet50 to VGG8.
> > >
> > > **4) Overstated claims: *"I  feel  that  some  claims  are  quite  bold,  e.g.,  ”This  enables  us to prove that  KD  using  only  pairwise  feature  kernel  comparisons  can  improve  NN performance  in  such  settings"***
> > > We thank the reviewer for pointing our this concern, as we are very keen to avoid overstating our claims. In this instance, we are simply stating that the result of Theorem 2 (factoring in Theorem 1) is theoretically showing that  there  is  a  provable  benefit  to  FKD  compared  to  standard  NN  training  in  the setting that we describe in Section 3.  This is not a  subjective claim  but  rather  an objective  mathematical  result,  which  we  prove  in  Appendix  C.  We  hope  that  this satisfies the reviewer that we are not being quite bold here, and we would appreciate if the reviewer can point out other claims of ours which they perceive to be quite bold.
> > >
> > > We hope that our rebuttal has allayed concerns that the reviewer had about our work, and that they may consider raising their score if so.

---

### Official Review · Reviewer_NGSC · 2021-10-29

**Correctness:** 4
**Technical Novelty And Significance:** 3
**Empirical Novelty And Significance:** 3
**Recommendation:** 8
**Confidence:** 3

**Main Review:**

This is an interesting paper. The paper is clearly written. Though I did not follow the detailed proof in the appendix, the brief description of the proof in the main paper was helpful to understand the idea behind the theorem.

The following are minor questions:
In eq(4), sum-pooling is used. In practice, however, max-pooling is also widely employed. Does the theoretical analysis also work for the max-pooling?
In numerical experiments, how was the regularization parameter lambda_KD determined?
When the genuine ReLU is used instead of tilde{ReLU}, can one obtain almost the same numerical results?


**Summary Of The Paper:**

There are several approaches in knowledge distillation for neural networks. This paper studies kernel-based knowledge distillation for multi-view data set. The authors revealed that the assembled feature map obtained from the ensemble teacher network effectively improves students' generalization ability. Some numerical results are presented to show the practical effectiveness of the learning method based on theoretical findings.


**Summary Of The Review:**

This is an interesting paper. The paper is clearly written. So my recommendation is to accept this paper.

---

> ### Author Response · Authors · 2021-11-15
> **Thank you! Initialial author response below**
>
> Thank you very much for your insightful feedback! Please find replies to points your raised below:
>
> **1) Max pooling: *"In practice, however, max-pooling is also widely employed.  Does the theoretical analysis also work for the max-pooling?"***
>
> Thank you for raising this interesting point, as we had not considered max-pooling before.  Yes the theoretical analysis will also extend to max-pooling and in fact the proofs will be simpler than for sum-pooling.  The intuitive explanation why is that, at the moment with sum-pooling, we show that features are learnt by showing that the sum of activations over patches when an attribute $v_{c,l}$ is present is ‘large’.  With max pooling, we will not need to worry about the ‘weaker’ patches  where  the  attribute  is  not  present  potentially ‘cancelling  out’  the  stronger patches where the attribute is present.  In terms of our proofs, max-pooling means we do not need to show that equations (14) & (15) are small, and can focus on equation (13) on its own.
>
> **2) Experiments: *"In numerical experiments, how was the regularization parameter $\lambda_{KD}$ determined?"***
> $\lambda_{KD}$ (along with all hyperparameters) was determined using 60 iterations of Bayesian optimisation (BO), optimising for validation accuracy on a validation set of size 10% of the training data.  Using our optimal hyperparameters, we retrained on the whole training data set to produce the results reported in the paper. For the BO,we used a uniform prior $\lambda_{KD}∼\text{Unif}([30,3000])$ over two orders of magnitude.
>
> In the rebuttal phase, we have run an ablation regarding the effect of $\lambda_{KD}$ for the VGG13$\rightarrow$VGG8 distillation experiment for FKD on CIFAR100. Test accuracies (%) for different $\lambda_{KD}$ are shown (mean+2std over 3 runs):
> - $\lambda_{KD}=10$: $71.34\pm0.09$
> - $\lambda_{KD}=30$: $71.88\pm 0.12$
> - $\lambda_{KD}=100$: $72.96\pm0.11$
> - $\lambda_{KD}=300$: $73.98\pm0.25$
> - $\lambda_{KD}=1000$: $72.37\pm0.32$
> - $\lambda_{KD}=3000$: $69.19\pm 0.53$
>
> We have also run the same ablation for other teacher/student architectures, and found a consistent trend where $\lambda_{KD}\approx300$ is optimal. We will add these to a revised version of appendix.
>
> **3) ReLU vs tildeReLU: *"When the genuine ReLU is used instead of tildeReLU, can one obtain almost the same numerical results?"***
> All experiments we ran used the genuine ReLU, whilst the tildeReLU was for theoretical ease due to its continuous monotonic gradients

---

### Official Review · Reviewer_68WG · 2021-11-02

**Correctness:** 3
**Technical Novelty And Significance:** 3
**Empirical Novelty And Significance:** 2
**Recommendation:** 6
**Confidence:** 3

**Main Review:**

The problem investigated in this paper is an interesting one. The approach of trying to regularise to match the kernels of the teacher rather than temperature scaled logits is also quite interesting. The experimental results shows that the method is competitive in various settings. The theory is interesting though it seems to be a fairly straightforward extension of Allen-Zhu and Li.

The one weakness of the paper is the lack of concordance between the theory and the practice. The experiments show that FKD is quite competitive even without ensembles. But the theory does not explained why. It was not clear to me how Feature Regularisation would effect the theory.

**Summary Of The Paper:**

In this paper the authors consider extending the knowledge distillation (KD) framework to cases where the student and the teacher do not share prediction spaces. To this end they propose Feature Kernel Distillation (FKD), where one views the last layer weights of a neural network as a data dependent feature kernel. Instead of regularising the distance between temperature scaled logits corresponding to each data point of the student and the teacher in vanilla Knowledge Distillation, the authors instead regularise based on the distance between feature kernels of a pair of data points (inner product of weights of the pre-final layer of the NN).

Under a simplified setting, viewing data as being multi-view the authors show a setting where classical knowledge distillation would not do well, but a FKD based knowledge distillation using an ensemble of teachers does well (the teacher kernel between two data points is the average teacher kernel across the ensemble). The authors then propose using a correlation kernel instead of raw kernels and feature regularisation to spread out the kernel values.

The authors proceed to empirically demonstrate  that this approach helps in dataset transfer between datasets of different number of classes. They also show that using an ensemble of teachers enables FKD to improve performance (but KD also improves performance).

**Summary Of The Review:**

The paper examines an interesting problem. The Feature Kernel Distillation is an interesting algorithm. The experiments seem to indicate that it is competitive in a variety of settings. The theory is reasonable interesting as well. The paper is well written as well. Hence I recommend acceptance.

---

> ### Author Response · Authors · 2021-11-15
> **Thank you! Initial author response below.**
>
> Thank you very much for your considered and constructive feedback! Below we address points raised:
>
> **1) Clarification: *Under a simplified setting, viewing data as being multi-view the authors show a settingwhere  classical  knowledge  distillation  would  not  do  well,  but  a  FKD  based  knowledge distillation  using  an  ensemble  of  teachers  does  well  (the  teacher  kernel  between  two data  points  is  the  average  teacher  kernel  across  the  ensemble)."***
> We  do  not  wish  to overstate our contributions and so we would like to point out that we do not show that classical knowledge distillation (Hinton et al 2015) would fail in our setup.  Having said that, (echoed in our response to Reviewer DKJu), it would be straightforward to construct a setting where FKD would work fine but it would not be clear how to apply vanilla KD as the teacher and student have different output dimensions or where the classes are randomly permuted between student and teacher datasets.
>
> **2) Theoretical novelty: *"...fairly straightforward extension of Allen-Zhu and Li."*** Despite our response to point 1 above, we would like to respectfully contest that our theory is a fairly straightforward extension of Allen-Zhu and Li, whose work we believe we extend in a considerable way. Please can we refer to the discussion regarding point 1) of Reviewer DKJu’s author response for further clarification.
>
> **3) Theory vs Experiments: *" The one weakness of the paper is the lack of concordance between the theory  and  the  practice.   The  experiments  show  that  FKD  is  quite  competitive  even without  ensembles.   But  the  theory  does  not  explained  why."***
> We  believe  there  is  a misunderstanding here.  Our theory does show that FKD works well without ensembles, because one can think of a single teacher as an ensemble of size $E=1$.  Then Theorem 2  still  shows  an  advantage  of  using  a  single  teacher, $E= 1$,  for  FKD  compared  to standard supervised learning training which corresponds to $E= 0$.
>
> **4) Feature regularisation: "*It was not clear to me how Feature Regularisation would effect the theory"***
>
> We believe the feature regularisation aspect of FKD is motivated by a combination of our approach of aiming to ensure that the student’s feature kernel is well suited for generalisation after the distillation process (which sets us apart from previous ‘similarity’ distillation approaches such as Relational KD (Park et al, 2019) or Similarity Preserving (Tung and Mori, 2019) that aimed purely at transfering knowledge from theteacher to student), as well as our theory. In particular, our use of correlation kernels to avoid learning noise (detailed in appendix D.1), and thus a desire to keep control over the norms of the feature kernel over different inputs (Figures 3 and 7).  This is as opposed to the feature regularisation impacting our theory itself, though this would be certainly an interesting area to explore further.  We would like to also point out Appendix F.2 which describes a negative hypothesis we had about the motivation for feature regularisation.  Outside of the KD literature, feature regularisation has shown to match performance on (then) SOTA EfficientNet model on ImageNet in Dauphin & Cubuk (2021), so a theoretical understanding of feature regularisation is pertinent future work beyond FKD and the scope of this work. Finally, we would like to add that even without Feature Regularisation FKD outperforms the related SP & RKD methods and Vanilla KD (Hinton et al 2015), as depicted in Tables 1 and 2.

---

### Official Review · Reviewer_DKJu · 2021-11-04

**Correctness:** 3
**Technical Novelty And Significance:** 2
**Empirical Novelty And Significance:** 2
**Recommendation:** 6
**Confidence:** 4

**Main Review:**

Pros:

- The proposed FKD is novel, to my knowledge. Especially, it can apply to the setting where the teacher's output dimension is different from that of the student, while the traditional KD is not applicable.
- The authors provided theoretical justifications for their FKD under a carefully designed setup, which I believe intuitively makes sense.
- The entire paper is well-organized and the authors provided detailed and intuitive explanations when they introduce the theorems and derivations, which makes it easier for me to follow the paper.

Cons:

- The technical novelty is very limited. I roughly went through the appendix, and I found most of the derivations are based on the results in  Allen-Zhu & Li (2020). There were no novel proof techniques presented.

- It's unclear for me, how does FKD compares to KD theoretically. If I understand correctly, FKD, KD, and ensembles are all able to deal with the multi-view setting, where standard training will fail to generalize. So, what's the advantage of FKD over KD in theory? I think some clarifications on this are essential.

- Why there is an $\epsilon$ in theorem 2? I don't understand where it comes from. Can you explain?


Minors:

In Definition 2 in Appendix C.1, page 18 & the second equation in Appendix C.4, page 27, $\mathcal{P}\_{v, l}$ should be $\mathcal{P}\_{v_{c,l}}$.


Overall, I believe the proposed FKD is a novel and (potentially) practically-useful algorithm. However, I think the theoretical results in the paper do not provide any new insights on KD. So, I would suggest the authors move the theoretical results into the appendix, and focus on providing more empirical results, e.g., NLP tasks. I believe this will better sell the paper, and also expose the paper to a broader audience.

**Summary Of The Paper:**

This paper proposed a new knowledge distillation framework, feature kernel distillation (FKD), by matching the student network's feature kernel and that of the teacher network's feature kernel. The authors extended the theoretical results in Allen-Zhu & Li (2020) to demonstrate the advantages of their framework over standard NN training. The theoretical results are built upon a synthetic dataset consisting of multi-view and single-view data, where standard training will fail to generalize while FKD will generalize well. Empirically, the authors (loosely) showed the consistency between the empirical behavior of FKD and the theoretical predictions, though the empirical setting is fairly different from the theoretical setting. In addition, on several image datasets, FKD can outperform other KD baselines in terms of test accuracy.

**Summary Of The Review:**

I like the idea of FKD that matches the student network's feature kernel with that of the teacher network in KD. This can apply to situations where the teacher network and student network have different output dimensions. However, the theoretical novelty is limited, and also the empirical evaluations are not thorough (only on some very small datasets). In summary, I vote for a weak acceptance.

---

> ### Author Response · Authors · 2021-11-15
> **Thank you! Initial author response below. [1/n] technical novelty discussion**
>
> Thank you for your thoughtful and encouraging review! Please find our replies below:
>
> **1) Theory: *"the technical novelty is very limited. I roughly went through the appendix,...There were no novel proof techniques presented*"**
>
> We respectfully disagree that our technical novelty is ‘very limited’. We believe we build on the results of Allen-Zhu & Li (2020) in a substantial and practically meaningful way by considering feature kernels as distillation objects, as follows:
>
> (a) There are several related works which consider using notions of ‘similarity’ or ‘relatedness’ of input pairs to distill information from teacher to student (please see our response to Reviewer h8ud). To the best of our knowledge, none of these related works had theoretical justification, and so our work can be seen as providing a theoretical motivation for many of these works: we believe our work is an important technical contribution which validates that these previous methods, which are used in practice, are principled. This isn't to say that FKD is not novel methodologically: we include a detailed discussion of differences to previous related works in Appendix D and moreover our experiments demonstrate that FKD consistently outperforms related works (due to our theoretically motivated practical considerations in section 3.1).
>
> (b) FKD is fundamentally different to vanilla KD, as the former learns features by encouraging the student to match the teacher in how similar to view *pairs* of images, whereas the latter learns features by comparing how likely student and teacher think a *single* image belongs to different classes. This fundamental difference is reflected throughout our proofs in Appendix C relative to those of Allen-Zhu & Li (2020).
>
> (c) Moreover, it is not clear from Allen-Zhu & Li (2020) (who work with vanilla KD only and do not consider feature kernels) what the appropriate definition of feature kernel even is, and this is a subtle matter for CNNs (which our theory deals with) as to whether one (i) flattens the final layer representation $h(x)\in\mathbb{R}^{CmP}$ (known as vectorised feature kernel) or (ii) average/sum (known as Global Average Pooling, or GAP, feature kernel) pool over patches to give $h(x)\in\mathbb{R}^{Cm}$. Following the comments of Reviewer NGSC, we realised it is also possible to consider max pooling in our theory, which is similar to item (ii) above. The importance of this difference between (i) & (ii) is that the former does not allow intra-patch correlations whereas the latter does, which is discussed in more depth in the infinite-channel NNGP scenario in Novak et al (2018). The implications of this distinction is that the former will only register large feature kernel values for inputs which share the same attribute which are located in the same patch. Using our intuitive cat eyes/car headlights analogy, this would require the headlights of a car to be located in exactly the same part of the image as the eyes of a cat to register a large feature kernel value. Recall the number of patches $P$ are $O(C^2)$ where $C$ is number of classes, whereas the number of patches per attribute is a finite constant, and so it is possible to construct a setting where it is very unlikely for two images to share attributes in the same patch and hence the vectorised feature kernel would fail whereas our choice of GAP feature kernel succeeds as proved in Theorem 2.
>
> (d) Our results (Theorem 2, & in Appendix C) have an explicit dependence on the finite ensemble size $E$, showing that generalisation performance improves with $\frac{1}{2^{E+1}}\mu$, whereas Allen-Zhu & Li (2020) only consider either a large ensemble, $E=\text{polylog}(C)$ (Theorem 3 of Allen-Zhu & Li (2020)), or an ensemble of size $E= 1$ (Theorem 4 of Allen-Zhu & Li (2020)), for class size $C$ (we apologise for the clash of notation with Appendix C).
>
> (e) There are several other novel objects in the appendix that we consider which are necessary to analyse for Theorem 2, which one does not need in Allen-Zhu & Li (2020), such as $\Upsilon_{c,l}$ in Definition 4 or $\mathcal{M}_\mathcal{T}$ in Appendix C.2.
>
> (f) We believe, in addition to our mathematical theory, our technical contribution also includes using our feature kernel based approach to knowledge distillation to motivate implementation considerations, which we show work in practice in Section 4. Our motivation is not just to use the teacher’s feature kernel for knowledge distillation, but rather to try to make the student’s feature kernel as well suited for improved generalisation as possible, using both knowledge from the teacher and also an understanding of the student’s training process. This distinction, which separates FKD from other feature kernel based methods (as discussed in detail in Appendix D), also motivates our use of feature regularisation for the student in Section 3.1, which we show leads to a significant improvement in Tables 1 & 2.

---

> > ### Author Response · Authors · 2021-11-15
> > **[2/n] Theory continued**
> >
> > **2) Theory: "*it’s unclear for me, how does FKD compares to KD theoretically. If I understand correctly, FKD, KD, and  ensembles  are  all  able  to  deal  with  the  multi-view setting,  where  standard  training  will  fail  to  generalize.   So,  what’s  the  advantage  of FKD over KD in theory?  I think some clarifications on this are essential*"**
> >
> > We believe that the ability of FKD to apply over outputs of different sizes is a significant theoretical advantage over vanilla KD in and of itself.  It would be simple to construct a theoretical situation where say the teacher is trained on $C$ classes but the student’s dataset has $C+1$ classes. It would be relatively straightforward to adapt Theorem 2 to ensure that FKD would improve student performance in such a setting, whereas it is really not clear how to even apply vanilla KD, let alone prove theoretically that it works.  Another even simpler setting is to consider a setting where the student has exactly  the  same  classes  as  the  teacher  (cat=class  0,  car=class  1,  dog=class  2  etc) but with a randomly unknown permuted class numbering (cat=class 1, car=class 2, dog=class 0 etc).  FKD in Theorem 2 is unaffected by the permutation, whereas it is easy to see that vanilla KD will fail in the setting of Allen-Zhu and Li (2020).  We also highlight that FKD does not require a temperature hyperparameter (which needs tuning) unlike vanilla FKD, which we mentioned in Section 3.1.
> >
> > **3) Theory:  "*Why there is an $\epsilon$ in theorem 2?"***
> > The $\epsilon$ is included to account for the fact that for any independent individual trained member of the ensemble $e$,  with trained parameters $\theta_e^{\*}$ , we know that (following the notation of Appendix C.2),  $C \geq |\mathcal{M_e}| \geq C(1-o_p(1))$. Intuitively, the explanation is that it is possible (albeit with very low probability that goes to 0 as $C,m\rightarrow\infty$), for neither of the two attributes $v_{c,1},v_{c,2}$ for a given class to  be  correlated  with  the  initialised  parameters  (e.g.   it  is  technically  possible  for all initialised parameters to be negatively correlated with $v_{c,1}$,$v_{c,2}$) meaning that no neuron for class $c$ is ever activated throughout training), so that neither attribute is learnt by $\theta^{\*}_e$. The $\epsilon$ we introduce is to guard against this edge case in Theorem 2, whilst maintaining the key message that FKD improves student performance and moreover is better with larger ensemble size.

---

> > > ### Author Response · Authors · 2021-11-15
> > > **[3/3] NLP and Automatic Speech Recognition (ASR) experiments**
> > >
> > > **4) Experiments: "*So, I would  suggest  the  authors  move  the  theoretical  results  into  the appendix, and focus on providing more empirical results, e.g., NLP tasks"***
> > >
> > > In the rebuttal period, we performed the following proof-of-concept analyses to show effectiveness of our proposed framework and FKD in different tasks:
> > >
> > > a) **NLP**: We performed analyses for a neural machine translation (NMT) task proposed by Tan et al.  [NLP-1].  In the analyses, we could only obtain data for En-De (from English to German) translation since links to the datasets for other languagesare broken.  Therefore, we employed a self-distillation method on a pre-trained English model for En-De translation as follows:
> > >
> > > (i) We train a single teacher transformer model T on the IWSLT dataset for English [NLP-1].
> > >
> > > (ii) We perform self-distillation on T for En-De translation [NLP-1].
> > >
> > > (iii)  We  did  not  search  for  optimal  hyperparameters,  and  used  default  parameters  of the code provided by the authors of [NLP-1].  The results are as follows (accuracy is computed by BLEU):
> > >
> > > 1. En - De Individual [NLP-1]:  27.32
> > >
> > > 2. En - De Self-distillation [NLP-1]:  27.49
> > >
> > > 3. En - De Self-distillation (Hinton et al., 2015):  27.51
> > >
> > > 4. En - De Self-distillation [FKD]:  27.64
> > >
> > > 5. En - De Self-distillation (Replacing distillation loss (2) of [NLP-1] with FKD): 27.79
> > >
> > > We first note that, we adapted vanilla KD and our FKD for sequential data in the KD loss (2) of [NLP-1] in this task.  More precisely:
> > > - We first computed vanilla KD and FKD on token probabilities, and added these loss functions  to  the  KD  loss  (eq 2  of  [NLP-1])  in  (3)  and  (4).   In  the  results,  aggregating vanilla KD with the KD loss (eq 2 of [NLP-1]) improved accuracy from (2) 27.49 to (3) 27.51.  However, FKD further boosted BLEU to (4) 27.64.
> > > [NLP-1] Tan et al., Multilingual Neural Machine Translation with Knowledge Distillation, ICLR 2019.
> > > -  We then replaced KD loss (eq 2 of [NLP-1]) with FKD for training.  Remarkably, FKD further boosted the BLEU to (5) 27.79.
> > >
> > > These results suggest that the proposed FKD can be applied in NMT tasks, successfully.  We hope that these results will motivate researchers to employ FKD in various different NLP tasks including but not limited to multilingual NMT, NER and Q&A.
> > >
> > > b) **ASR**: We performed an analysis using a CRDNN model (VGG + LSTM,GRU,LiGRU+ DNN) on the TIMIT dataset.  In this experiment, we used a distillation approach proposed in [ASR-2] for ASR tasks as follows:
> > >
> > > (i) We train a single teacher model T on the TIMIT dataset [ASR-1].
> > >
> > > (ii) We perform self-distillation on T [ASR-2].
> > >
> > > (iii) We did not search for optimal hyperparameters, and used default parameters of the SpeechBrain Library.
> > >
> > > (iv) In this task, replacing CTC/NLL distillation losses with KD (Hinton et al., 2015) did not converge.  Additional investigation with hyperparameter search is needed.  We used phoneme error rate (PER) to measure accuracy of models.
> > >
> > > The results are as follows (lower is better):
> > >
> > > 1. Teacher:  Valid PER = 13.26
> > > 2. Distilled Teacher [ASR-2]:  Valid PER = 12.80
> > > 3. KD (Hinton et al., 2015):  Valid PER = 12.86
> > > 4. FKD:  Valid PER = 12.59
> > >
> > > Similar to the NMT tasks, we adapted vanilla KD and our FKD for sequential data as follows:
> > > - We first computed vanilla KD and FKD loss functions on token probabilities, and then added to the total loss (eq 7 of [ASR2]).
> > > - In the analyses, Vanilla KD (3) increased the PER from (2) 12.80 to (3) 12.86.
> > > - However, FKD further improved the PER from (2) 12.80 to (4) 12.59.
> > > -  In  this  task,  training  models  by  replacing  CTC/NLL  distillation  losses  (eq 4  or 5 of  [ASR2] ) with  KD  (Hinton  et  al.,  2015)  and  FKD  did  not  converge.   Additional investigation with hyperparameter search and theoretical analyses for distillation onsequential data and ASR models may further improve the accuracy.
> > >
> > > In  conclusion,  these  results  propound  that  FKD  can  be  applied  for  different  tasks, i.e.,  image  classification,  NMT  and  ASR,  boosting  accuracy  of  baseline  distillation methods.   We  hope  that  these  initial  results  will  motivate  researchers  in  different communities (computer vision, NLP, and ASR) to further expound and apply FKD in additional sub-tasks.
> > >
> > > [NLP-1] Tan et al., Multilingual Neural Machine Translation with Knowledge Distillation, ICLR 2019
> > >
> > > [ASR-1] Ravanelli et al., SpeechBrain:  A General-Purpose Speech Toolkit, 2021.
> > >
> > > [ASR-2] Gao et al., Distilling Knowledge from Ensembles of Acoustic Models for Joint CTC-Attention End-to-End Speech Recognition, 2021

---

### Author Response · Authors · 2021-11-15
**New experiments**

Thank you to all reviewers for their time and effort, we really appreciate it! We would like to draw to all reviewers' attention additional experiments that we have conducted in the rebuttal period, which show the performance of FKD in settings with more classes and beyond image classification. To avoid repeating ourselves, we add in brackets the individual reviewer rebuttal where the experimental results may be found:

1. NLP and ASR experiments (Reviewer DKJu, response 3/3)
2. ImageNet and Tiny-Imagenet transfer (Reviewer h8ud, response 2/3)
3. Regularisation strength ablation (Reviewer NGSC)
4. Comparison to Probabilistic Knowledge Transfer (PKT) (Passalis and Tefas 2018) on transfer experiments (Reviewer h8ud, response 1/3)

We will add these new results to a revised version of our submission.

---

### Decision · Program_Chairs · 2022-01-20

**Decision:**

Accept (Poster)

**Comment:**

This is a borderline paper.
This paper proposed feature kernel distillation (FKD), a new distillation framework, by matching the kernels obtained from the networks of student and the  teacher.  Theoretical justification is provided by  extending the results of Allen-Zhu and Li(2020)(ALi20 hereafter). Empirical results show superiority of FKD over vanilla KD on several datasets.
There is however concern that the technical novelty is limited and  incremental, an opinion shared by DKJu, and 68WG, compared to ALi20. Reviewer DKJu suggests that the authors could highlight those results which are not straightforward extensions of ALi20. Another important point of concern is that the paper may have some Overstated claims.  The authors clarified that the language of the claims be suitably edited. In this regard Reviewer h8ud have some specific suggestions which should be easy to incorporate.

In view of additional experiments conducted and detailed discussion during rebuttal addressed some of the concerns of the reviewers.
If accepted, the final version, should include most of the discussion and additional experiments.